# Interpretable and integrative analysis of single-cell multiomics with scMKL
Samuel D. Kupp [1], Ian A. VanGordon Jr[1], Mehmet Gönen [2,3], Sadık Esener [1,4], Sebnem Ece Eksi [1,4,5] & Çiğdem Ak [1,4] ✉

The rapid advancement of single-cell technologies has led to the development of various analysis methods, each with trade-offs between predictive power and interpretability particularly for multimodal data integration. Complex machine learning models achieve high accuracy, but they often lack transparency, while simpler models are more interpretable but less effective for prediction. In this manuscript, we introduce an innovative method for single-cell analysis using Multiple Kernel Learning (scMKL), that merges the predictive capabilities of complex models with the interpretability of linear approaches, aimed at providing actionable insights from single-cell multiomics data. scMKL excels at classifying healthy and cancerous cell populations across multiple cancer types, utilizing data from single-cell RNA sequencing, ATAC sequencing, and 10x Multiome. It outperforms existing methods while delivering interpretable results that identify key transcriptomic and epigenetic features, as well as multimodal pathways– that existing methods have failed to achieve, in breast, lymphatic, prostate, and lung cancers. Leveraging insights from one dataset to inform analysis in a new dataset, scMKL uncovers biological pathways that distinguish treatment responses in breast cancer, low-grade from high-grade prostate tumors, and subtypes in lung cancer, thereby enhancing our understanding of cancer biology and tumor progression.

Multiomic single-cell technologies simultaneously capture transcriptional and epigenomic states at the level of individual cells, transforming our understanding of complex biological processes in cancer biology. However, there is a critical need for computational methods that can assess the incremental contributions of each modality and facilitate the transfer of insights across datasets to accelerate discoveries in these complex systems. Currently, multiomic analysis frameworks, such as MOFA + [1] are an extension of multiomic factor analysis that was originally developed for bulk multiomics[2], while the Seurat/Signac suite[3,4] was initially designed to integrate multiple omics from different cells. These methods require extensive data processing, including normalization and dimensionality reduction, which distorts and underestimates biological variation, particularly in high-expression genes[5].

Recent efforts have shifted toward deep learning methodologies, specifically autoencoders[6–9] which capture non-linear structure by mapping high-dimensional data into a low-dimensional latent space. While effective for cell clustering and survival analysis[9], their latent representations are difficult to interpret, often requiring post-hoc explanations, that introduces bias, opacity, and ethical concerns[10].

To address these challenges, we developed scMKL, a scalable, and inherently interpretable machine learning approach for integrative analysis of single-cell multiomics data. scMKL combines multiple kernel learning (MKL) with random Fourier features (RFF) and group Lasso (GL) formulation, enabling transparent and joint modeling of transcriptomic (RNA) and epigenomic (ATAC) modalities.

scMKL groups features of single-cells to capture biological mechanisms, recognizing that genes do not act independently but as part of a group within their biological context. It leverages prior expert knowledge, such as pathways for RNA and transcription factor binding sites (TFBS) for ATAC. Instead of the conventional approach of first selecting a subset of gene expression features (usually 2000 most variable features[3,4]), then utilizing them in machine learning algorithms and finally, performing extensive downstream analysis as a second tool for interpretation; scMKL combines these steps and finds underlying cross-modal interactions between transcriptomics and epigenomics that opaque methods fail to capture.

scMKL supports unimodal (scRNA-seq or scATAC-seq) or multimodal (RNA + ATAC) inputs, and outputs interpretable model weights of each feature group in classification. We used Hallmark gene sets from

[1]Cancer Early Detection Advanced Research (CEDAR), Knight Cancer Institute, OHSU, Portland, OR, USA. [2]Department of Industrial Engineering, College of Engineering, Koç University, İstanbul, Türkiye. [3]School of Medicine, Koç University, İstanbul, Türkiye. [4]Department of Biomedical Engineering, School of Medicine, OHSU, Portland, OR, USA. [5]Division of Oncological Sciences, Knight Cancer Institute, OHSU, Portland, OR, USA. ✉e-mail: ak@ohsu.edu

Molecular Signature Database and TFBS from JASPAR and Cistrome databases as prior biological knowledge to guide kernel construction[11–13]. Instead of relying on post-hoc explanations, scMKL directly identifies the regulatory programs and pathways driving cell state distinctions.

We benchmarked scMKL against both supervised and unsupervised state-of-the-art methods across seven datasets (Table 1), including four cancer types (breast, prostate, lymphoma, and lung) and three sequencing technologies (scRNA-seq, scATAC-seq, and multiome), demonstrating its superior accuracy, robustness, scalability, and interpretability. We first tested scMKL on a breast cancer cell line with two conditions where the cell labels and the interpretation were straightforward (i.e., connections between molecular features and biological functions are well-understood), allowing us to accurately assess prediction and interpretation abilities. scMKL accurately identified key regulatory pathways and TFs involved in the estrogen response (ER). Then we validated the learned insights on an independent breast cancer cell line experiment, showcasing scMKL's potential in transfer learning. We demonstrated scMKL's generalizability using sciATAC-seq data differentiating low- vs. high-grade prostate tumors from multiple patients, revealing tumor subtype-specific signaling mechanisms. We benchmarked scMKL on challenging classification tasks using two independent non-small cell lung cancer (NSCLC) scRNA-seq datasets collected under distinct protocols to evaluate performance and biological relevance across cohort shifts, subtype comparisons, and class imbalance scenarios.

Together, scMKL encapsulates a scalable, integrative model that improves classification accuracy and enables interpretable multimodal analysis, supporting hypothesis generation across multiple disease states and facilitating both basic science and translational discoveries.

## Results
### Overview of the scMKL approach
scMKL is the first MKL framework designed to integrate RNA and ATAC data at the single-cell level, overcoming key scalability and interpretability limitations of traditional kernel-based approaches. While MKL methods have been applied to bulk multiomics datasets like TCGA[14,15], they operate at the patient level and are not scalable to single-cell resolution and interpretable at the feature or modality level.

Few kernel-based methods exist for single-cell RNA-seq analysis, including SIMLR[16] and the spectral clustering approach[17], but these are limited to small datasets, do not support multimodal integration, and offer limited biological interpretability. Others, like PIMKL[18], employ pathway-induced kernels for bulk RNA data but cannot scale to single-cell or incorporate epigenomic modalities like ATAC.

scMKL addresses these gaps through innovations in both computational efficiency and biological interpretability. It uses RFF to reduce complexity from $O(N^2)$ to $O(N)$, and GL regularization for sparse, modality-aware feature selection. This design scales to large, high-dimensional datasets while identifying informative feature groups across modalities.

Unlike prior methods, scMKL constructs omic-specific kernels aligned with the structure of RNA and ATAC data, integrates expert-curated biological knowledge, and provides interpretable weights reflecting the biological signals driving classification (Fig. 1a). Prior knowledge for grouping features is summarized in Table 2.

To avoid overfitting and enhance reproducibility, scMKL repeated 80/20 train-test split 100 times with cross-validation to optimize the regularization parameter λ (Fig. 1b). Stronger regularization (higher λ) results in fewer selected pathways ($\eta_i \neq 0$), increasing model sparsity and interpretability while reducing potential overfitting. Conversely, lower λ values allow the model to capture more biological variation but may compromise generalizability (Fig. 1c).

We applied scMKL to a diverse set of multiome and single-omic cancer datasets (Table 1), including multiome (RNA + ATAC) and single-omic scRNA-seq and scATAC-seq profiles from MCF-7 and T-47D breast cancer cell lines, small lymphatic lymphoma (SLL), and prostate cancer (PCa), and two non-small cell lung cancer (NSCLC) subtypes: lung adenocarcinoma (LUAD), and lung Squamous cell carcinoma (LUSC). The MCF-7 and T-47D multiome datasets focus on the impact of estrogen treatment as compared to controls, while the SLL multiome dataset investigates healthy vs. tumor states in patient samples. The PCa datasets include independent scRNA-seq and scATAC-seq data, differentiating non-malignant vs. malignant, and Gleason 3 vs. Gleason 4 grade tumors, respectively. LUAD and LUSC scRNA-seq investigate healthy vs. cancer cells, respectively. These datasets represent diverse cancer subtypes, stages, and platforms, supporting evaluation of scMKL's generalizability.

We benchmarked scMKL on three major tasks: (i) predicting ER in MCF-7 cells, a well-annotated, well-established interpretable setting used to assess both accuracy and pathway selection; (ii) stratifying low- vs. high-grade PCa cells using sciATAC-seq data; (iii) evaluating scMKL on a set of challenging transfer learning tasks, leveraging two independent NSCLC datasets that differ in collection time and computational processing pipelines. These analyses demonstrate scMKL's accuracy, flexibility and capacity to uncover critical insights into disease progression and subtyping.

### scMKL outperforms state-of-the-art classification algorithms
We first benchmarked scMKL against standard kernel-based classifiers, including Support Vector Machines (SVM) and EasyMKL. While SVM uses a single kernel, EasyMKL supports multiple kernels but is limited in scalability. On the smallest dataset, scMKL achieved superior classification accuracy while training 7× faster and using 12× less memory than EasyMKL (Fig. 2d). Although we matched the number of pathway-informed kernels for fairness, existing methods became computationally impractical at larger scales (Fig. 2c, d).

We also compared scMKL to three widely used classification algorithms, Multi-Layer Perceptron (MLP)[19,20], Extreme Gradient Boosting (XGBoost)[21–23], and SVM, across three single-cell multiome (RNA +

## Table 1 | Summary of Datasets

| Dataset | # Cells | # RNA Features | # ATAC Features | # GAS Features | Classification Task | Class Sizes |
|---|---|---|---|---|---|---|
| MCF-7 (multiome) | 6438 | 36,601 | 206,167 | - | Control vs. Estrogen Treated | 2995 / 3443 |
| T-47D (multiome) | 6615 | 36,601 | 206,167 | - | Control vs. Estrogen Treated | 3028 / 3587 |
| SLL (multiome) | 13,498 | 36,601 | 109,789 | - | Healthy vs. Tumor | 10,587 / 2911 |
| PCa (scRNA) | 13,322 | 21,877 | - | - | Non-malignant vs. Malignant | 8664 / 4658 |
| PCa (scATAC) | 12,717 | - | 560,578 | 55,442 | Gleason 3 vs. Gleason 4 | 7383 / 5334 |
| LUAD (scRNA) | 74,084 | 37,698 | - | - | Healthy vs. Cancer | 68,229 / 5855 |
| LUSC (scRNA) | 16,287 | 37,698 | - | - | Healthy vs. Cancer | 13,970 / 2317 |

This table provides an overview of the datasets utilized in scMKL analysis, including number of cells, feature counts for RNA, ATAC, and Gene Activity Score (GAS) data, and the corresponding classification tasks. For each dataset, the number of cells per class is also shown, with class labels indicating the biological comparisons (e.g., control vs. estrogen-treated, healthy vs. tumor, non-malignant vs. malignant). The datasets cover various conditions, including multiomics datasets (RNA + ATAC), single-omics datasets scRNA, and scATAC data, with diverse cancer types, facilitating a comprehensive evaluation of scMKL's performance across different omic layers and biological contexts.

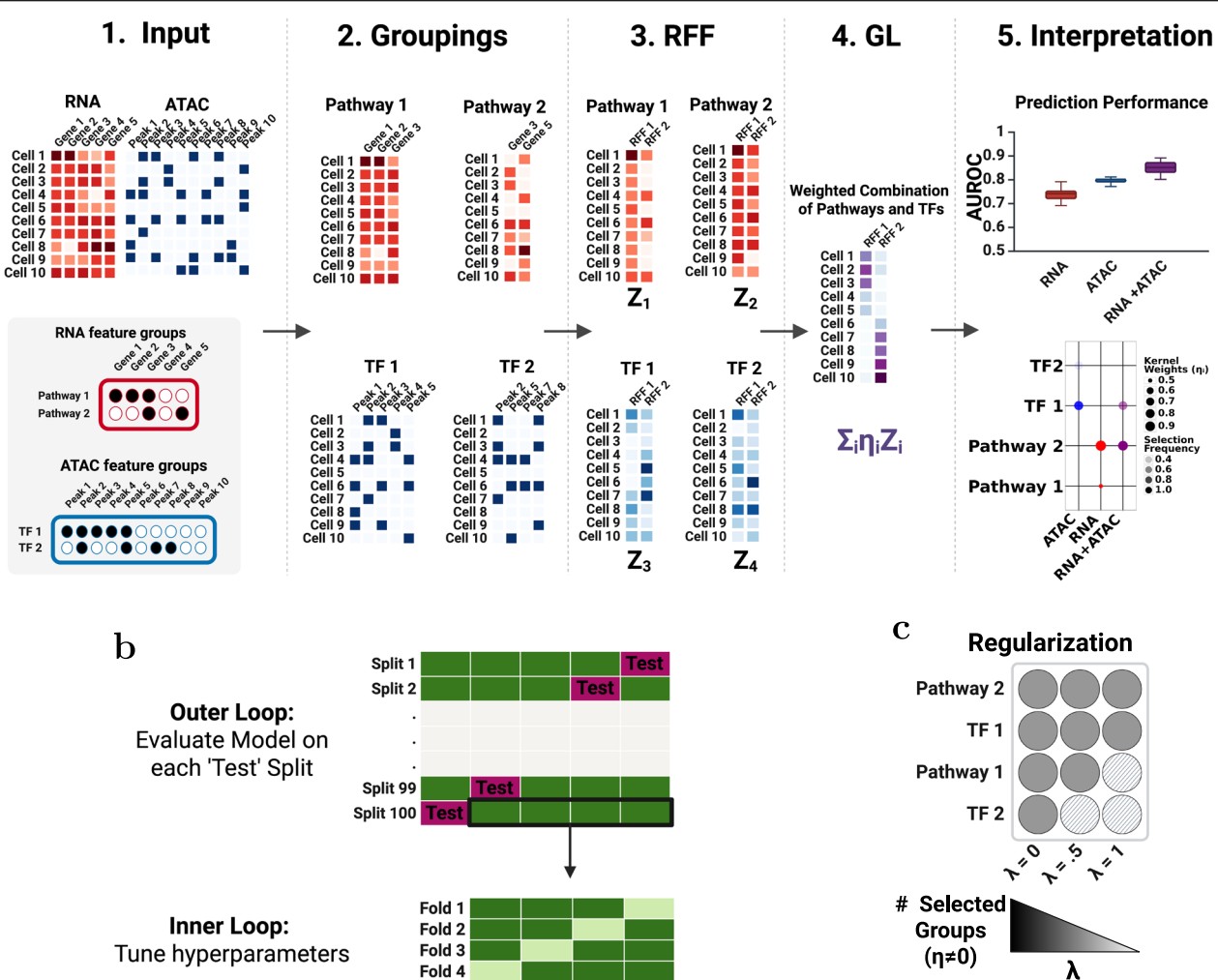

**Fig. 1 | Overview of scMKL: An interpretable machine learning framework for single-cell multiomics integration. a** *Multi-view and multi-modal integration in scMKL.* scMKL integrates single-cell RNA-seq and ATAC-seq data with expert-curated biological knowledge to enhance interpretability and performance. In the multi-view setting, modality-specific kernels are constructed from gene sets/pathways (RNA) and TFBS (ATAC), allowing distinct biological signals to be captured independently. Using the kernel approximations $Z_i$, GL identifies the optimal combination of biological signals, with the learned weights $\eta_i$ directly quantifying the contribution of each feature group to the classification outcome. Interpretability is visualized through dot plots showing selection frequency and weight of each pathway or TF group. **b** *Cross-validation and robust evaluation.* An outer loop using an 80/20 train-test split is repeated 100 times with different random seeds. Within each training set, 4-fold cross-validation is used to tune the regularization parameter λ. This nested setup ensures robust generalization and guards against overfitting. **c** *Effect of regularization on pathway sparsity.* Increasing λ leads to more sparsity in feature selection, reducing the number of pathways or TFs retained. This panel illustrates the trade-off between interpretability (fewer, more informative features) and potential biological mechanisms, underscoring the importance of tuning λ appropriately. See Table S1 for a full list of abbreviations used throughout the manuscript.

ATAC) datasets (MCF-7, T-47D, SLL) and four single-omic datasets: two PCa datasets (RNA, ATAC), and LUAD/LUSC (RNA) datasets. We assessed model performance using the area under the receiver operating characteristic curve (AUROC) as the primary metric across 100 independent models (Fig. 2a, b).

RNA-based models demonstrated consistently high AUROC values across all datasets, with scMKL outperforming all baselines and XGBoost showing the weakest performance (Fig. 2a, b, red boxes). We compared the performance of MLP, XGBoost, and SVM using all RNA features (~21–36K) vs. only Hallmark genes (~4 K). Contrary to scMKL, all other models performed best when utilizing all RNA features. SVM performed the worst overall in every comparison. Notably, scMKL achieved better results than all other algorithms statistically significantly ($p < 0.001$), despite using fewer genes (Fig. 2a, b).

For ATAC-based models, AUROC values varied more widely. We assessed the performance of MLP, XGBoost, and SVM using either the top 5000 most variable ATAC-peaks (ATAC MVF) or peaks overlapping Hallmark gene set regions (ATAC Hallmark) (Fig. 2b, blue boxes). We did not have the computational resources to run MLP, XGBoost, and SVM using all available ATAC features. We used the 5000 MVF as suggested in standard analysis as using all ATAC features did not efficiently scale in terms of memory and time. Standard analysis favored variable features, yet scMKL outperformed or matched all baselines when using biologically informed features. In MCF-7 and T-47D, prior-informed features outperformed MVF (Fig. 2b).

F1, precision, and recall scores (Fig. S1a, b) supported AUROC findings. While scMKL maintained strong performance across most datasets, its performance declined on two class-imbalanced datasets–LUAD and LUSC (class ratios of 12:1 and 7:1, respectively), which is explored further in the next sections. Due to the size of the LUAD dataset, it was not feasible to run all state-of-the-art methods without exceeding memory limits. To evaluate computational efficiency,

## Table 2 | Overview of RNA and ATAC Feature Groups

| Dataset | Number of Groups | Total Number of Features In All Groups | Number of Features per Group | Number of Exclusive Features to One Group | Number of Shared Features Across Groups |
|---|---|---|---|---|---|
| **Hallmark RNA** | | | | | |
| MCF-7 (multiome), T-47D (multiome), SLL (multiome), PCa (scRNA), LUAD (scRNA), LUSC (scRNA) | 50 | 4384 | 32–200 | 2671 | 1713 |
| **Hallmark ATAC** | | | | | |
| MCF-7 (multiome), T-47D (multiome) | 50 | 8335 | 75–540 | 4741 | 3594 |
| SLL (multiome) | 50 | 6407 | 38–416 | 3609 | 2798 |
| PCa (scATAC) | 50 | 12,208 | 95–591 | 7196 | 5012 |
| **Hallmark GAS** | | | | | |
| PCa (scATAC) | 50 | 4384 | 32–200 | 2671 | 1713 |
| **Cistrome ATAC** | | | | | |
| MCF-7 (multiome) | 121 | 186,537 | 642–76,604 | 19,683 | 166,854 |
| T-47D (multiome) | 20 | 152,103 | 1331–101,890 | 42,591 | 109,512 |
| **JASPAR ATAC** | | | | | |
| MCF-7 (multiome) | 720 | 199,838 | 938–2000 | 14,358 | 185,480 |
| T-47D (multiome) | 720 | 203,239 | 938–2000 | 8311 | 194,928 |
| SLL (multiome) | 720 | 109,560 | 640–2000 | 597 | 108,963 |
| PCa (scATAC) | 720 | 491,687 | 2000 | 127,115 | 364,572 |

This table provides a summary of the RNA and ATAC feature databases used in scMKL analysis, including Hallmark gene sets and TFBS from Cistrome and JASPAR. For each dataset, the table lists the number of biological groups, the total number of features that are in the groupings, the number of features per group, the number of features exclusive to one group, and the number of shared features across groups. Exclusive features refer to features found in only one grouping, while shared features refer to features found in multiple groupings. The datasets include multiomics and single-omic single-cell data from various cancer types, allowing for comprehensive analysis of molecular pathways, gene expression, and regulatory elements. The feature group databases support the interpretation of scMKL's performance across different biological contexts and omic layers.

we benchmarked runtime and memory usage with upper bounds of 12 h and 50 GB (Figs. 2c, d and S2a, b).

We compared the scalability of scMKL by testing performance on randomly sampled cell subsets of increasing size. scMKL provided the best AUROC for large sample sizes (Fig. 2c, d) while scaling comparably or better than MLP, XGBoost, and SVM in terms of time and memory (Fig. S2a).

We assessed how dimensionality reduction and preprocessing impact model performance. For ATAC data, only SVM performance improved with LSI but required approximately 18× more memory and 2× more compute time (Fig. S2b), while PCA on RNA data did not improve predictive accuracy. We benchmarked scMKL across preprocessing workflows (PCA, log-normalization, raw counts, TF-IDF, LSI, binary counts, and standard single-cell preprocessing) and found it consistently robust, achieving high AUROC (0.9731–0.9999) across datasets. No single preprocessing technique proved universally superior, underscoring scMKL's adaptability to preprocessing strategies. (Fig. S3).

Together, these advances establish scMKL as an efficient and biologically grounded interpretation tool without sacrificing performance, offering a practical advantage in resource-limited settings for multimodal single-cell data integration.

### TFBS-informed peak groupings improved performance
We investigated chromatin region groupings based on TF associations using two databases: JASPAR and Cistrome. We compared these to peak groupings derived from the Hallmark gene set (Fig. 2e). Across all datasets, JASPAR TF groupings produced higher AUROC than Hallmark, improving median AUROC by ~2% (MCF-7, T-47D), ~1% (SLL), and ~4% (PCa). Conversely, Cistrome TF groupings decreased performance for MCF-7 (0.4%) and T-47D (8%) despite being tissue-specific.

We also assessed whether the binary nature of ATAC-seq data impacted performance. We applied TF-IDF, commonly used in scATAC-seq data workflows[24], which improved AUROC by 1-2% for MCF-7 and T-47D with both Hallmark and JASPAR-informed peak groups (Figs. 2e, S4b). For other datasets, TF-IDF decreased accuracy (Fig. 2e). JASPAR and

Hallmark binary peak groups also had overall higher F1, precision, and recall (Fig. S2c).

To examine whether the accuracy stemmed from biologically meaningful prior information, we perturbed the feature groups by randomizing a portion of pathway features or added noise features instead in a second experiment (Fig. 2f, g). As we progressively increased the randomness or added noise within the feature groups, respectively, AUROC values decreased as expected in each case. This decline was most evident for RNA Hallmark, TF-IDF ATAC Hallmark, binary ATAC Cistrome, and Binary ATAC JASPAR groupings, suggesting these strategies are particularly effective in capturing meaningful biological mechanisms for MCF-7.

### Multimodal classification increases performance by combining complementary views
We integrated RNA and ATAC models to enhance accuracy and reveal interactions between modalities. Kernel functions are a natural method for multiomics datasets given their ability to integrate different data types. When we combine kernels from different modalities, we take advantage of the unique perspective offered by each. To our knowledge, no other methods integrate distinct modalities in this way, leveraging the unique insights provided by each and capturing complex cross-modal interactions, while avoiding the high dimensionality and information loss associated with other approaches[5,10].

Combining multiple data modalities provided higher prediction accuracy (Figs. 3a and S5a, c). For all datasets, the most accurate classification results were achieved by combining RNA Hallmark with ATAC JASPAR- with statistically significant improvements in all the metrics (Fig. S4).

### scMKL identifies multimodal pathways and cross-modal interactions that underlie the treatment response
To evaluate scMKL's interpretability, we analyzed estrogen-treated vs. control MCF-7 breast cancer cells using single-cell multiome (RNA + ATAC) data and different biological priors (Hallmark, Cistrome, and

**Fig. 2 | scMKL provides scalable, flexible, and accurate predictions across diverse datasets. a,b** *Classification performance comparison:* MLP, XGBoost, SVM, and scMKL were evaluated across seven datasets, including three multimodal datasets–MCF-7, T-47D, and SLL; and four unimodal datasets–PCa, LUAD, and LUSC scRNA-seq and PCa scATAC-seq. Darker color shades represent models trained on all features, while lighter shades denote models using Hallmark or MVF features for RNA and ATAC. *, **, *** are used to denote statistical significance from Wilcoxon test *p* values < 0.05, 0.01, and 0.001, respectively. Gold indicates scMKL had significantly better performance; black indicates the benchmark algorithm had significantly higher performance. **c** *Comparison of scalability cost:* Analysis of training time in seconds and memory in GB across different sample sizes (1k-6k cells) for the four machine learning models (MLP, XGBoost, SVM and scMKL) using all genes, Hallmark genes, Hallmark peaks, and most variable peaks. **d** *Comparison of MKL methods (EasyMKL vs. scMKL),* showing AUROC, memory usage, and runtime as a function of sample size. **e** *Comparison of ATAC data groupings and transformations:* Predictive performance for binary and TF-IDF normalized ATAC peaks using Hallmark gene sets, Cistrome, and JASPAR TFBS as prior information. *, **, and *** denote significance determined by Wilcoxon tests for *p* values < 0.05, 0.01, and 0.001 indicating which ATAC data transformation resulted in significantly better AUROC. **f, g**. Systematic perturbation analysis across RNA and ATAC Hallmark and JASPAR models, measuring performance decline as feature groups were progressively substituted or noise was added to assess biological informativeness. Error bars show standard deviation.

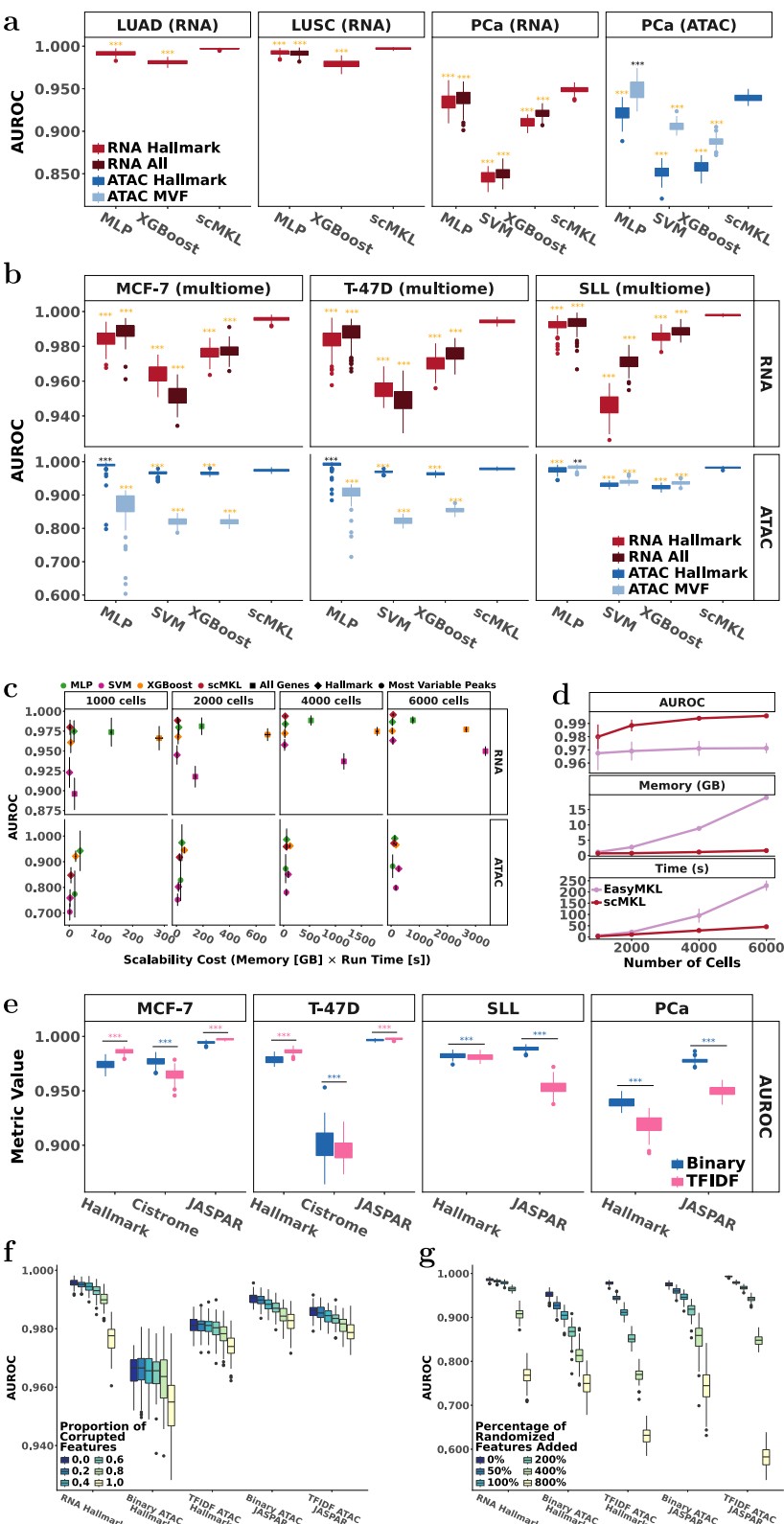

JASPAR). MCF-7 cells are from an estrogen receptor-positive (ER +) breast cancer cell line. Estrogen treatment typically activates ER signaling pathways, leading to changes in gene expression and chromatin accessibility associated with disease progression[25].

We assessed which feature groups scMKL consistently selected across different levels of model sparsity (i.e., number of feature groups scMKL used for prediction) (Fig. 3c). Using RNA, ATAC and combined RNA + ATAC inputs, we quantified the informativeness of each Hallmark pathway and JASPAR TF by the selection frequency and the kernel weight (Fig. 3b). Multimodal integration (RNA + ATAC) outperformed unimodal models (RNA, ATAC) across F1-score, precision, and recall (Fig. 3a). In the JASPAR-based ATAC data analysis, ESR1 and ESR2 were the top selected

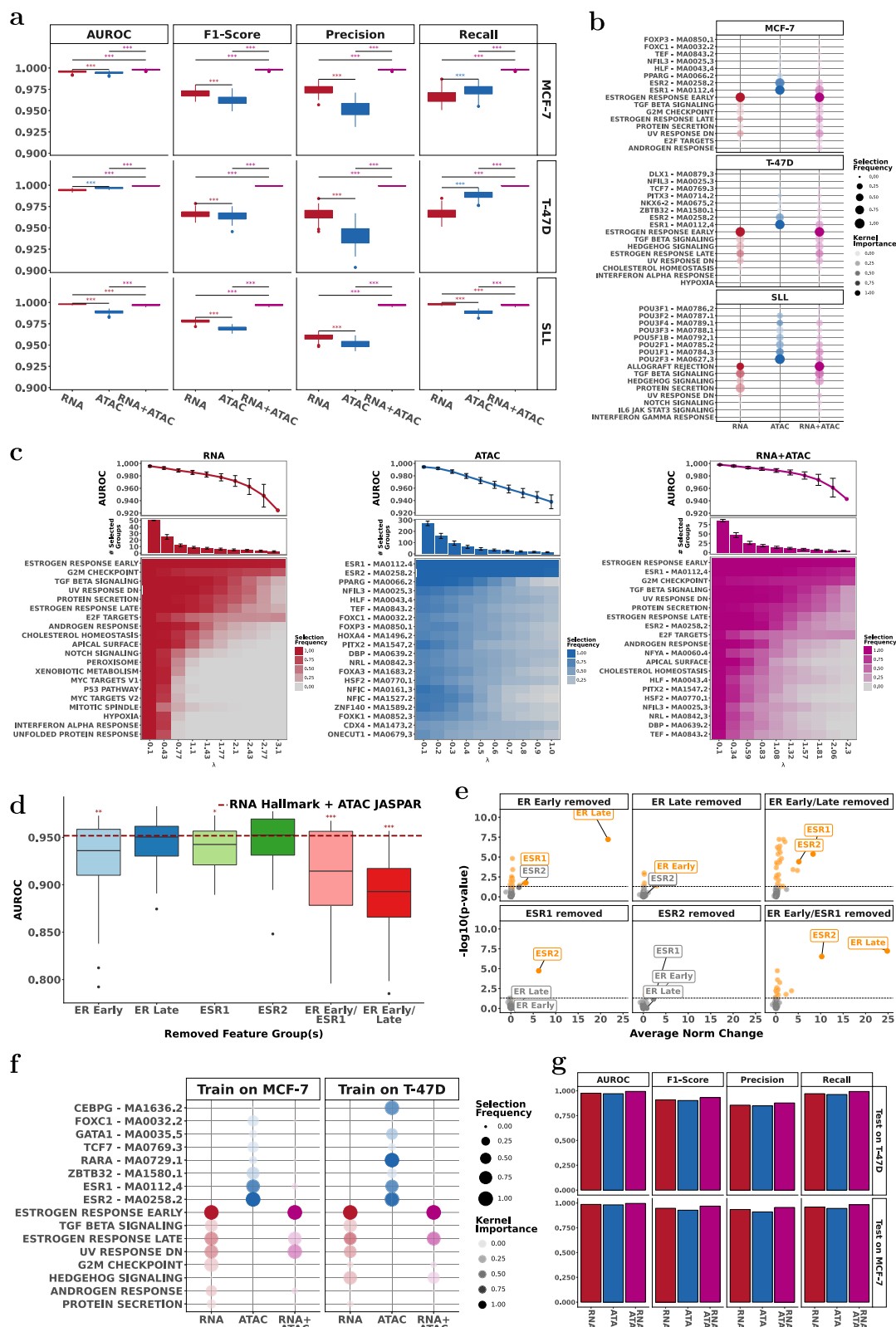

TFs (Fig. 3b, c). ESR1 (ERα) is the primary mediator of estrogen signaling in breast cancer, while ESR2 (ERβ) modulates ERα activity and may suppress tumors in certain contexts[25]. With RNA + ATAC, ER Early (RNA), and ESR1 (ATAC) groups were consistently prioritized, underscoring their importance in the ER. This consistency across modalities highlights scMKL's ability to detect robust, biologically meaningful signals.

To test the functional importance of top-selected feature groups, we performed ablation studies on MCF-7 by excluding top-ranked unimodal and multimodal feature groups from training; in all cases, model performance decreased, confirming that scMKL depends on biologically relevant features quality (Fig. 3d). Interestingly, when either ESR1/ESR2 or the ER Early/Late pathways were removed from training, scMKL adaptively up-

**Fig. 3 | scMKL enables accurate, interpretable multimodal classification and identifies key cross-modality biological mechanisms through robust feature selection. a** Comparison of classification performance (AUROC, F1-score, precision, recall) between RNA (red), ATAC (blue), and RNA + ATAC (purple) models across MCF-7, T-47D, and SLL multimodal datasets. *, **, and *** denote significance determined by Wilcoxon tests for $p$ values < 0.05, 0.01, and 0.001 respectively. The color indicates which experiment had significantly higher performance. **b** Feature group selection frequency and kernel importance scores across RNA, ATAC, and RNA + ATAC models, highlighting consistently selected biological pathways and TFs. **c** AUROC and feature group selection frequency across RNA, ATAC, and RNA + ATAC models, identifying critical feature groups consistently across different regularization values ($\lambda$). Error bars show standard deviation. **d,e** Ablation (leave-one-group-out) analysis showing the impact on AUROC and the importance of the other feature groups after removal of selected feature groups (e.g., ER Early/Late response, ESR1/ESR2) in the MCF-7 dataset compared to baseline RNA + ATAC model. *, **, and *** denote significance determined by Wilcoxon tests for $p$ values < 0.05, 0.01, and 0.001 indicating which ablation experiments resulted in significantly worse AUROC than the baseline. **f.g** Cross-dataset feature transferability: Selection frequencies and kernel importance when training on MCF-7 and evaluating on T-47D, and vice versa, along with corresponding model performance on the transfer learning task. Selection frequency and kernel importance values were averaged across sparsity parameter values, $\lambda$.

weighted the remaining estrogen-related features (Fig. 3e), indicating redundancy in ER signaling axis and the model's ability to re-prioritize biologically relevant groups.

Hallmark pathway selection across unimodal (RNA, ATAC) and multimodal (RNA + ATAC) settings showed consistent prioritization of ER Early and TGF-$\beta$ Signaling pathways, with RNA features dominating multimodal predictions (Fig. S5b). Figure S5e shows the co-selection of Hallmark pathways in the RNA unimodal prediction setting. Among the top 5 predictive pathways, both the ER Early and G2M pathways were chosen together, with the ER Early pathway consistently receiving higher weights. The ER Early pathway was most predictive, often selected alone with ~0.92 AUROC (Fig. 3c).

Cistrome analysis identified MED12 and SP1 as top TFs (Fig. S5c, d). MED12, a key regulator in multiple Hallmark pathways including TGF-$\beta$ signaling[24], remained prominent in RNA + ATAC models, frequently co-selected with ER Early, G2M Checkpoint and TGF-$\beta$ signaling pathways. Notably, the ER Early pathway consistently received higher weight and was most often paired with MED12 (Fig. S5f). We looked at the overlap of MED12-regulated regions with Hallmark pathway gene regions. This analysis confirmed that MED12 regions significantly intersected with ER Early pathway but not G2M checkpoint (Fig. S5g, h). These results highlight scMKL's ability to detect cross-modal interactions, linking regulatory factors and gene expression patterns, providing deeper insights into the mechanisms governing cellular processes.

### scMKL enables effective transfer learning across datasets
Next, we sought to use this gained insight on an unseen dataset. Two breast cancer cell lines, T-47D and MCF-7, with control vs. estrogen-treated cell groups were analyzed. We trained scMKL with one cell line and evaluated it on the other using the insight gained from the training (Fig. 3f, g). scMKL achieved high classification accuracy (AUROC > 0.96 per modality), despite the heterogeneity present between the cell lines (Fig. 3g). Though the relevant pathways and their weights were different between cell lines for ATAC; the key pathway for RNA and integrated RNA + ATAC was the same– the ER pathway. This highlights the power of scMKL to leverage knowledge from one dataset to accurately predict outcomes in another, making it a powerful tool for translating insights to patient data with similar biological complexity and variation. For this analysis, we did not perform bootstrapping; pathway selection dot plot reflects weights and selection frequency across ten $\lambda$ values (Fig. 3f).

### scMKL outperforms standard single-cell workflows in identifying biological mechanisms without extensive strict preprocessing
We compared our pathway selection to standard single-cell workflows (e.g., Seurat, scanpy), which include filtering, variable gene selection, dimensionality reduction, clustering, and differential expression analysis[26,27]. We also performed gene set enrichment using GSEApy[28].

Focusing on MCF-7 dataset, using scanpy we identified 5267 DEG, 1584 of which overlapped with Hallmark gene sets (Fig. 4a). GSEA ranked E2F targets highest, while scMKL prioritized ER Early, which ranked fifth in GSEA but was most predictive in classification (Fig. 4b). Even though only a few features are unique to each gene set, as shown in Fig. S6, scMKL

performed with high accuracy using the most relevant groupings, demonstrating the importance of grouping features in uncovering important biological insights rather than individual features alone.

We visualized UMAP embeddings based on the top 2000 most variable genes (scanpy), the top GSEA pathway (E2F targets), and the top scMKL pathway (ER Early) (Fig. 4d). Silhouette scores were 0.0503 (scanpy), 0.0474 (GSEA), and 0.0675 (scMKL), with ER Early yielding the clearest separation.

scMKL's most frequently selected genes, including *MYC*, *CCND1*, and *ELOVL5*, were identified based on pathway-level selection (Fig. 4c). Bar plots are color-coded based on the z-scores of DEG from the E2 group. Genes displayed in gray were not identified as DEGs but were included by scMKL due to their presence in frequently selected pathways. DEG analysis confirmed that the top three frequently selected genes by scMKL are DE between E2 and Vehicle cells. *MYC* and *CCND1* have been well-associated with cancer through many studies, and their respective proteins play important roles in the cell cycle[29,30]. *ELOVL5* has also been shown to be related to cancer, and estrogen exposure in MCF-7 cells has been linked to its up-regulation[31]. This shows that scMKL can be used to find genes related to treatment response despite the group-level focus.

Comparisons across other datasets are available via our interactive Shiny app: https://huggingface.co/spaces/scMKL-team/scMKL_analysis.

### scMKL identifies pathways that stratify low- and high-grade PCa
We applied scMKL to sciATAC-seq dataset derived from 18 localized prostate tumors graded as primary Gleason 3 (G3, low-grade) and Gleason 4 (G4, high-grade). Previous analysis with cisTopic, a topic-modeling-based unsupervised method, identified four cell clusters (Fig. 5h)[32]. The biggest cluster was from low-risk patients who were graded as G3, and cells in the other three clusters were from high-risk tumors that were graded as G4. Patient samples are inherently more heterogeneous than cell lines, consisting of several cell types and states in and around the tumor regions, often lacking clear ground truth labels. Despite tumor heterogeneity, scMKL classified tumor grade with AUROC > 0.95 while providing robust pathway-level interpretation (Fig. 5a, d).

When distinguishing G3 and G4 cells, scMKL frequently selected multiple pathways, including androgen response (AR), WNT, and Hedgehog signaling, reflecting the biological complexity of tumor samples (Fig. 5b, c). Individually, AR and WNT Signaling pathways achieved AUROC of 0.72 using Gene Activity Score (GAS), while Hedgehog, WNT, Notch Signaling pathways achieved AUROC of 0.68 using ATAC data; and AR, WNT, Peroxisome, Hedgehog Signaling pathways achieved an AUROC of 0.71 using ATAC and GAS combined (Fig. 5c). Interestingly, key PCa drivers, including WNT, Hedgehog, Jak-STAT, Notch, and AR, were selected more frequently when ATAC signals were combined with the GAS, underscoring the value of capturing cross-modal regulatory interactions.

To validate these results, we used an independent scRNA-seq dataset generated from prostate tumors with similar Gleason grade distribution[33] (Fig. 5f). Despite the differences in clinical features, methods, and data types, scMKL identified overlapping pathways, including AR, apical surface, angiogenesis pathways that separated malignant from non-malignant cells.

Next, we examined the frequency of TFs selected for classifying single-cells from primary G3 and G4 prostate tumors (Fig. 5d, e). Consistent with

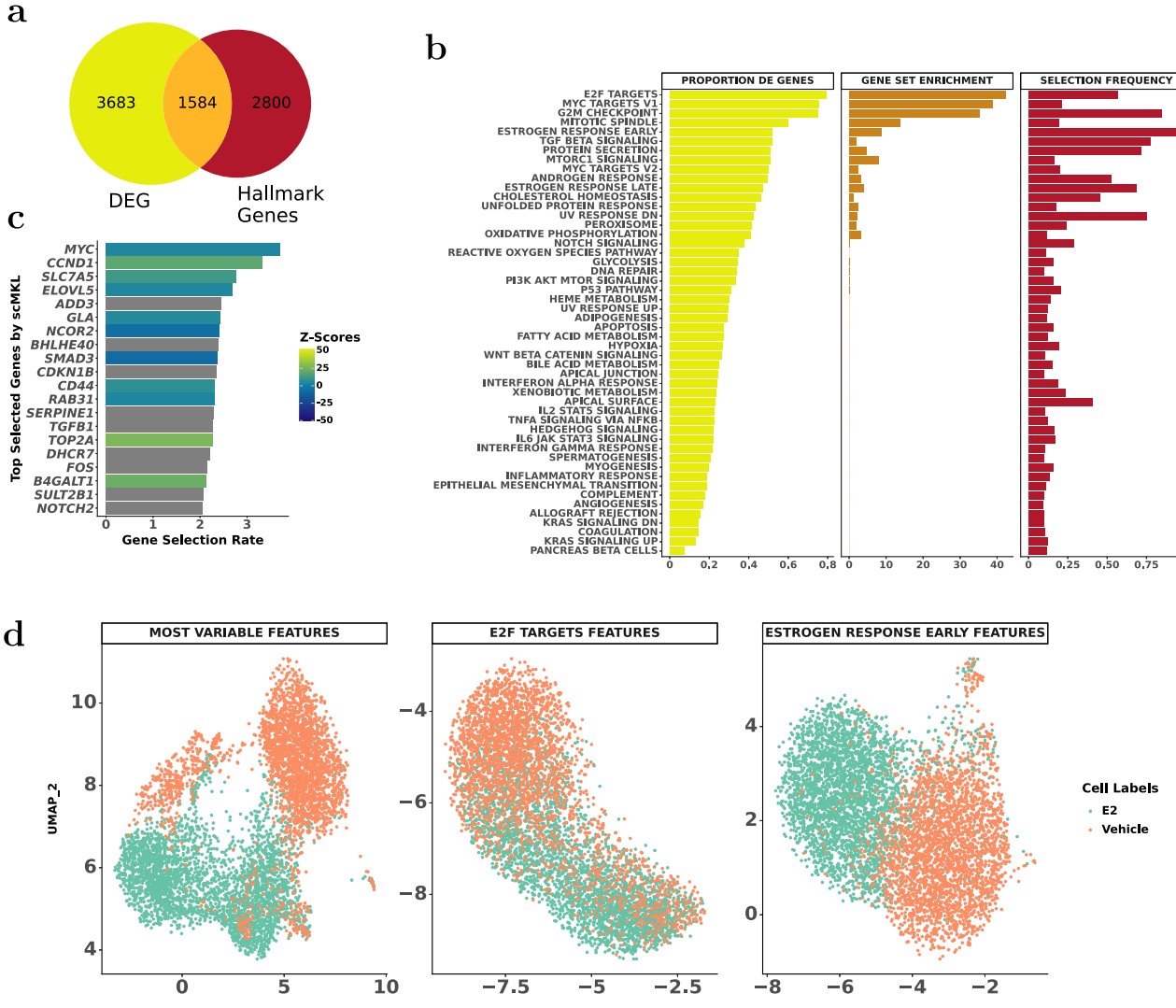

**Fig. 4 | scMKL more effectively identifies underlying biological mechanisms in MCF-7 cells compared to the current the state-of-the-art single-cell analysis methods. a** *Gene overlap analysis:* Overlap between genes in Hallmark gene sets and differentially expressed genes (DEGs) between E2-treated and Vehicle-treated MCF-7 cells. **b** *Comparison of scMKL with DEG overlap and gene set enrichment:* DEG overlap with Hallmark gene sets, Gene set enrichment from GSEApy for GO Biological Processes gene sets, and scMKL selection on MCF-7 cells. **c** *Gene selection by scMKL:* Top genes selected by scMKL for classifying MCF-7 cells, with selection frequency represented, and colored by their enrichment in E2-treated cells. **d** *UMAP visualization:* UMAP of MCF-7 cells using 2000 most variable genes, E2F target genes, and ER Early genes to depict cell clustering and treatment-specific expression patterns.

prior studies, canonical TFs, such as FOXA1, HOXB13, and CDX2 were among the top 50 features[33] (https://huggingface.co/spaces/scMKL-team/scMKL_analysis). In addition, our scMKL pipeline identified several TFs associated with neuroendocrine (NE) differentiation in PCa, including OTX2, TFDP1, and SREBP2. Both OTX2 and SREBP2 have been implicated in lineage plasticity of prostate tumors and are known regulators of C-MYC[34–36], while TFDP1 promotes the proliferation and is frequently upregulated advanced disease[37]. Together, these results suggest activation of lineage plasticity factors, such as OTX2 and SREBP2, is detectable in G4 tumors, occurs even before treatment with AR inhibitors such as enzalutamide, aligning with prior motif enrichment analyses while improving interpretability.

To refine our understanding of AR regulation and NE differentiation, we used the Labrecque gene signature, which classifies prostate tumors into five categories based on AR and NE gene expression[38]. Additionally, we developed expert-guided NE gene sets from the Beltran study[38,39] (Fig. 5g). This allowed us to further classify NE-specific transcriptional profiles and provided a more nuanced view of the signaling pathways that drive NE differentiation within prostate tumors. Using GAS alone, AR-related

pathways were most predictive for distinguishing G3 vs. G4 cells (Fig. 5g, red panel). When TF groupings were incorporated, four of the five NE pathways ranked among the top 25 most informative, indicating their significance in differentiating tumor grades (Fig. 5g).

Using the cisTopic UMAP on our sciATAC-seq dataset (Fig. 5h), we investigated NE pathway differences between individual G4 clusters (G4_1, G4_2, and G4_3) and the aggregated G3 cluster (Fig. 5i). While AR pathways from both Labrecque and Beltran were the most predictive for G4 aggregate, they were less frequently selected within individual G4 clusters. Conversely, the NEPC pathway was most predictive for G4-1; Labrecque Neuro I and II pathways were the most selected for G4-2 and G4-3 differentiation. Each cluster was classified with high AUROC values (0.75, 0.76, and 0.77, respectively), illustrating how scMKL captures patient-specific regulatory signatures within high-grade tumors.

## scMKL generalizes across cohorts and tumor subtypes while handling class imbalance

To evaluate scMKL's generalization across tumor subtypes, cohorts, and technical variation, we benchmarked its performance on six classification

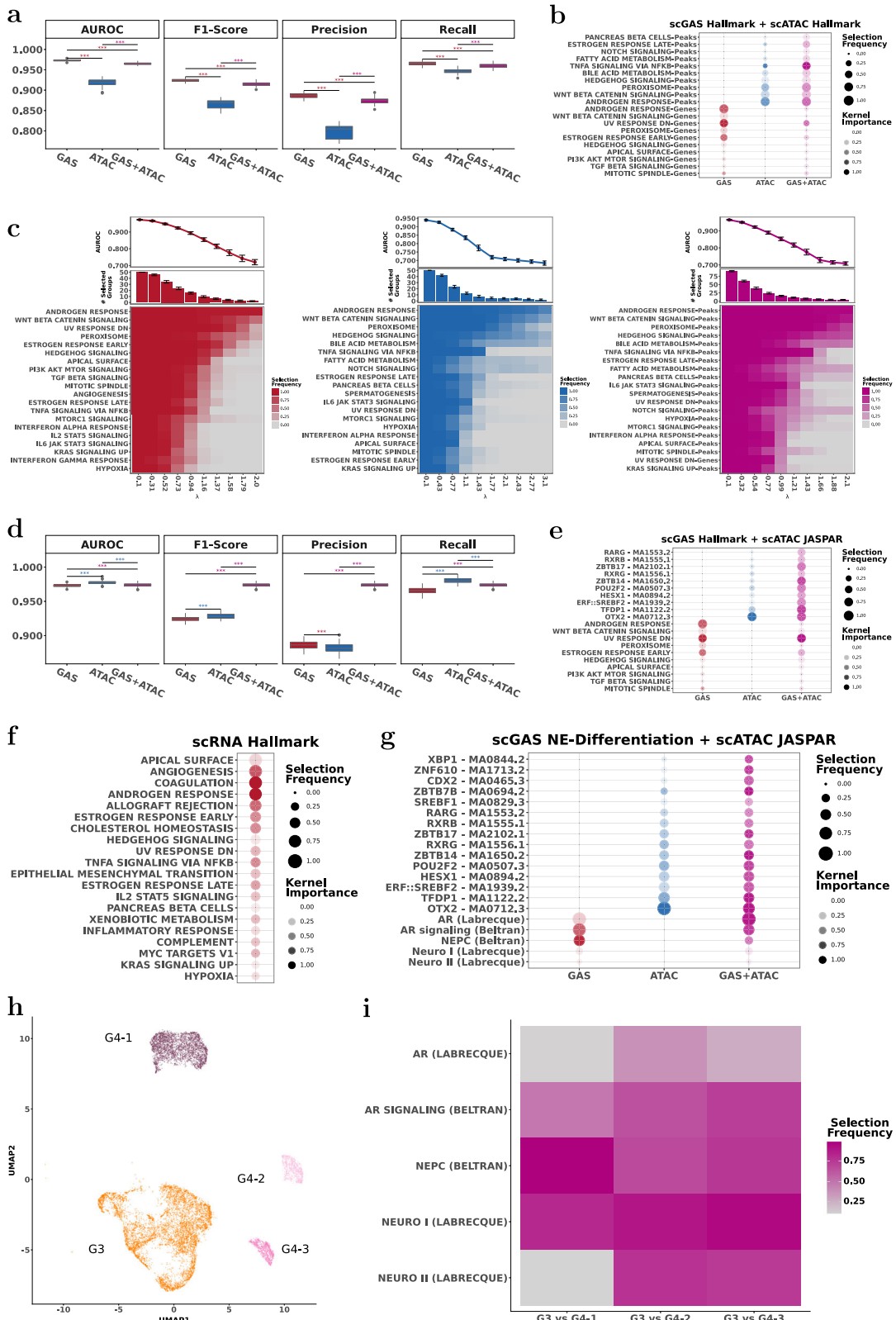

tasks using two independent NSCLC scRNA-seq datasets[40–42], which include samples from both LUAD and LUSC (Fig. 6a, c).

across all tasks. To address class imbalance, healthy cells were sub-sampled in the training set.

**Dataset construction and evaluation strategy.** We organized datasets by cancer subtype to generate separate LUAD and LUSC cohorts. scMKL was trained on Maroni et al. (2021)[41] and tested on Zilionis et al. (2019)[40]

**Classification tasks.**

1. Pooled healthy vs. cancer classification: distinguishing healthy vs. cancer cells using combined LUAD and LUSC cohorts.

**Fig. 5 | scMKL stratifies cells from low- and high-grade prostate cancer tumors, identifying factors associated with disease progression and subtypes. a,b** Classification performance (AUROC, F1-score, precision, recall) between GAS (red), ATAC (blue), and combined GAS + ATAC (purple) models in PCa scATAC-seq data. along with Hallmark pathway selection frequency and kernel importance scores, highlighting consistently selected biological pathways. *, **, and *** denote significance determined by Wilcoxon tests for p values < 0.05, 0.01, and 0.001 respectively. The color indicates which experiment had significantly higher performance. **c** AUROC as a function of the sparsity parameter λ (line plot) for scATAC PCa. Also shown are the top 20 selected pathways for each given λ. Shown for GAS Hallmark (red), ATAC Hallmark (blue), and GAS + ATAC Hallmark (purple). **d,e**

Classification performance (AUROC, F1-score, precision, recall) between GAS (red), ATAC (blue), and combined GAS + ATAC (purple) models in PCa scATAC-seq data, along with TF selection frequency and kernel importance scores, highlighting consistently selected biological pathways and TFs. **f** Hallmark pathway selection frequency and kernel importance scores using an independent scRNA-seq dataset classifying malignant vs non-malignant cells **g** Selection frequency and kernel importance scores of multimodal NE-differentiation gene sets and JASPAR TFs in classification of G3 vs G4 cells. **h** The Cistopic UMAP with clusters colored by grade, orange = G3, three shades of pink = G4−1, −2, and −3 clusters. **i** NE differentiation pathway selection frequency in classification of G3 cells vs individual G4 clusters. Error bars show standard deviation.

2. LUAD-specific classification: distinguishing healthy vs. cancer cells within LUAD cohort.
3. LUSC-specific classification: distinguishing healthy vs. cancer cells within LUSC cohort.
4. Tumor subtype classification: distinguishing LUAD tumor cells vs. LUSC tumor cells.
5. Cross-subtype generalization I: training on LUSC and testing on LUAD (healthy vs. cancer).
6. Cross-subtype generalization II: training on LUAD and testing on LUSC (healthy vs. cancer).

**Robustness to class imbalance**. We varied the healthy-to-cancer ratios and evaluated classification performance across ten replicates (Fig. 6b). scMKL remained stable across a wide imbalance range (Figs. 6b, S7a). Performance improved with greater feature group inclusion (lower λ) under skewed ratios, while moderate sparsity (higher λ) performed best near balanced ratios (0.6–1.6).

TNFα signaling via NF-κB and the Reactive Oxygen Species (ROS) pathways were consistently co-selected across all transfer learning experiments (Fig. 6e). These are tightly interconnected pathways; TNF-α is known to induce ROS production in lung cancer cells, promoting oxidative DNA damage and contributing to tumor progression[43,44]. Their repeated selection underscores a shared inflammatory and stress-response axis across NSCLC subtypes. Recent reviews underscore the therapeutic potential of modulating ROS levels as a strategy to eliminate cancer cells by exploiting their metabolic vulnerabilities[44].

Allograft rejection was identified in every case except the LUAD vs. LUSC comparison (Case 4), consistent with its role in distinguishing broad immune activation, thus more relevant for distinguishing tumor from non-tumor cells, rather than between tumor subtypes.

The ER Late pathway was consistently selected in LUAD-related comparisons (Cases 1, 2, 4, and 6), but not in LUSC-specific settings (Cases 3 and 5), pointing to LUAD subtype specificity. Supporting this, survival analysis in TCGA revealed a significant association between ER Late pathway activity and patient outcome in LUAD, while no such association was observed in LUSC (p < 0.01 in LUAD; non-significant in LUSC, Fig. S7b, c). These results are consistent with prior studies implicating estrogen receptor signaling in LUAD biology[45] and treatment response[46], and suggest this pathway may have prognostic and therapeutic relevance in LUAD. The KRAS signaling pathway followed a similar pattern, further supporting LUAD-specific regulation, which shows that KRAS mutations have a higher prevalence in LUAD[47,48].

MYC targets were consistently selected in LUSC and in LUAD–LUSC subtype comparisons, but were notably absent in LUAD-only classification, indicating its stronger association with LUSC biology and potential as a distinguishing feature. In line with this, previous work has shown that MYC protein expression was elevated in LUSC tumors relative to LUAD tumors, and MYC overexpression induced a "squamous-like" phenotype[49].

Finally, when comparing tumor subtypes (Case 4), scMKL identified additional pathways related to epithelial–mesenchymal transition (EMT), interferon-alpha response, and metabolism, suggesting that inter-subtype

classification tasks capture broader transcriptional and regulatory variation[50]. These results illustrate scMKL's ability to mine biologically meaningful and context-specific signals, supporting its utility for mechanistic hypothesis generation and experimental prioritization.

## Discussion

We developed scMKL, a computational method for analyzing single-cell multiomics data with built-in interpretability and the ability to integrate heterogeneous modalities. Unlike traditional black-box models, scMKL incorporates prior knowledge, including curated pathways and regulatory elements, into its learning process and captures cross-modal interactions. By combining RFF, MKL, and group-lasso regularization, scMKL delivers precise and transparent predictions grounded in decades of genomic research[11,13].

As high-dimensional omics data continues to accumulate, integrating all data modalities and combining all the insights gained from these different modalities into a unified model is likely to be key to providing high-quality, targeted bioinformatics analysis. scMKL outperformed MLP, XGBoost, and SVM across all datasets with less variance and lower resource usage (Figs. 2 and S1). With Hallmark+JASPAR multimodal integration, scMKL achieved statistically significant improvements across all datasets and evaluation metrics, consistently outperforming ATAC-only baseline methods (Fig. S4).

In heterogeneous cancer datasets, such as single-cell data from prostate and lung cancer patients, which inherently exhibit significant heterogeneity, scMKL consistently identified relevant biological pathways, which enhances its reliability in various experimental conditions, including the selection of similar pathways across multiple train-test splits (Figs. 5c, 6e). We also showed scMKL's flexibility and its robustness to variations in preprocessing steps (Fig. S3).

Beyond accuracy, scMKL is computationally efficient (Figs. 2c, d, S2a, b). Its design allows scaling to large-scale single-cell studies while maintaining interpretability. scMKL achieves this due in part to its efficient kernel design and group-aware sparsity constraints. We benchmarked runtime and memory usage under varying feature-group sizes, revealing that scMKL remains scalable and efficient even with large prior sets, such as JASPAR TFs (Fig. S8).

In contrast to standard single-cell workflows, which require downstream differential expression, enrichment analysis, and additional interpretation tools, scMKL replaces the need for post hoc interpretation tools (Fig. 4). It adjusts its interpretation when key features are ablated. For example, removing ESR1 or its pathway from training caused compensatory upweighting of ESR2 or related ER pathways, demonstrating scMKL's biologically coherent feature prioritization (Fig. 3d, e).

To test generalizability, we applied scMKL to class imbalance and cross-cohort prediction tasks in NSCLC. Across six transfer learning scenarios, including cross-subtype classification (LUAD/LUSC) and cohort transfer, scMKL maintained high performance (Fig. 6). scMKL identified subtype-specific pathways that align with well-characterized mechanisms in NSCLC. These results were consistent across datasets collected at different times and processed with different computational pipelines. We identified biologically relevant, subtype-specific pathways, such as ER in LUAD and MYC signaling in LUSC, which could inform drug sensitivity profiling or

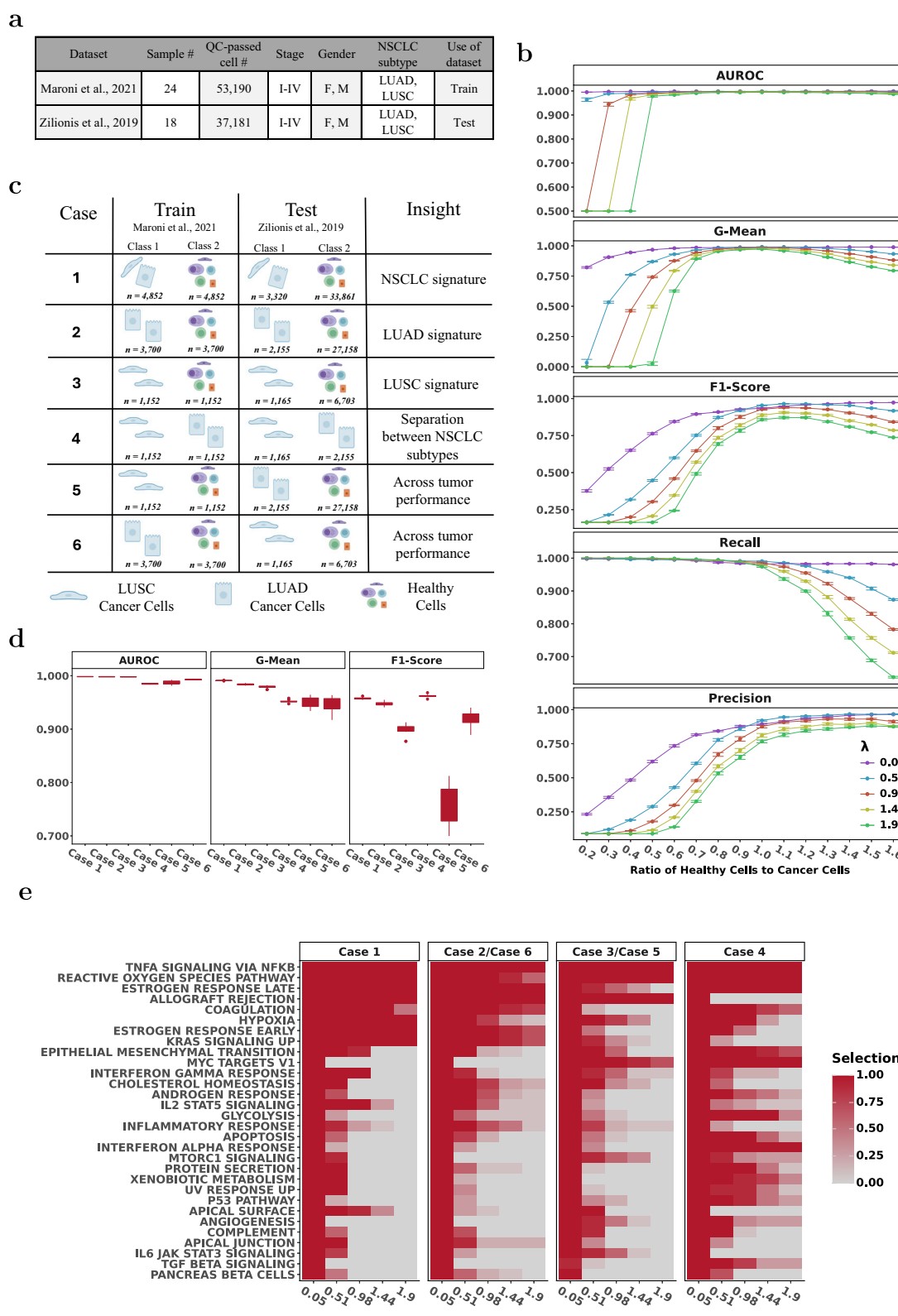

**Fig. 6 | scMKL demonstrates robust generalization under transfer learning and class imbalance settings in NSCLC. a** Overview of the NSCLC cohorts used for training and testing. **b** Impact of class imbalance on classification performance across metrics (AUROC, G-mean, F1-score, recall, and precision) under varying healthy-to-cancer cell ratios and regularization strengths (λ). Error bars show standard deviation. **c** Experimental setup for 6 classification tasks spanning subtype-specific and cross-cohort transfer learning scenarios. **d** Classification performance of scMKL across all tasks, showing strong generalization between LUAD and LUSC subtypes and across cohorts. **e** Selected features across tasks, revealing subtype-specific signals, such as MYC signaling in cross-subtype settings and ER Late in LUAD-trained models.

patient stratification. These findings highlight the robustness of scMKL across biological and technical variation, including dataset shifts, tumor heterogeneity, and sample imbalance. By leveraging biologically informed kernels, scMKL uncovered features associated with patient outcomes, supporting hypothesis generation, therapeutic insight, and prioritization of experimental follow-up or biomarker discovery.

While we focused on RNA and ATAC data, scMKL is adaptable to any omics whose features can be linked to biological processes based on prior knowledge, such as DNA methylation or proteomics. For example, gene sets, TFBS, or protein interaction modules can serve as structured feature groups. Future directions include adapting scMKL for automated cell-type classification using cell-type-specific feature groupings, allowing reproducible and interpretable labeling of new datasets using existing cell atlases, and moving beyond traditional clustering and DEG-based methods. Its interpretability may facilitate the identification of cell types that are not represented in the training dataset.

A key limitation of scMKL is its dependence on prior information quality. If prior sets are poorly curated or irrelevant to the biological system under study, interpretability and accuracy may suffer. To mitigate this, pathway corruption analysis can help identify uninformative priors by measuring their alignment with classification outcomes (Fig. 2f, g). Yet, biological databases contain a wealth of information spanning genes[11], proteins[51], as well as cell-types and tissues[52]. By harnessing more knowledge and data modalities that complement each other, scMKL enables the generation of more detailed and interpretable representations of biological processes. When no prior information is available, for instance, a new disease, novel cell population, or an unknown treatment response, unsupervised approaches like topic modeling as an alternative method to identify feature sets specific to cell groups, generating data-driven prior information.

Ultimately, scMKL integrates prior knowledge and multimodal data into a single interpretable model that provides both predictive power and biological insights. This combination supports hypothesis generation, experimental prioritization, and deeper understanding of disease mechanisms, paving the way for more personalized and mechanistically grounded, effective therapeutic strategies in cancer and beyond.

## Methods
### Datasets
We used multiomic datasets consisting of RNA and ATAC modalities from breast cancer and lymphoma. We also used four unimodal datasets: scRNA-seq and scATAC-seq datasets from two different PCa studies and scRNA-seq datasets from two different NSCLC studies.

**10x multiome breast cancer cell lines.** MCF-7 and T-47D are breast cancer cell lines sequenced using the 10x Multiome platform[53]. cellranger-arc (10x Genomics, v2.0.0) was used for initial data processing. For each dataset, a subset of cells was exposed to 20 nmol/l $\beta$-estradiol for 24 h and the control and estrogen-treated groups were sequenced. In classification experiments, we used estrogen-treated (E2) vs control (Vehicle) as ground truth cell labels.

**10x multiome SLL.** SLL consists of a flash-frozen intra-abdominal lymph node tumor sequenced using 10x Multiome platform and publicly available by 10X. cellranger-arc (10× Genomics, v2.0.0) was used for initial data processing. Tumor cells were determined using clustering and cell markers from gene expression data. In classification experiments, we used tumor vs. non-tumor cell labels.

**sciATAC-seq PCa.** The data consists of 18 flash frozen tumors sequenced by single-cell combinatorial indexing ATAC-seq (sci-ATAC-seq)[32]. Initial processing was done in snapATAC and aligned using BWA. Samples have overall Gleason Scores identified by a pathologist, but cell labels were assigned computationally. In Eksi et al (2021), the cells were clustered using the top 30 topics from Cistopic, and then assigned cell-

types based on a Genomic Regions Enrichments of Annotations Tool analysis between clusters[32]. All cell-types except epithelial were removed and clusters of cells were labeled as Gleason Pattern 3 (G3) and Gleason Pattern 4 (G4). We repeated the Cistopic analysis and found the same cell clusters as previously found (Fig. 5h). Because this dataset does not have simultaneous measurement of gene expression, we used the GAS calculated in the previous study as a second omic. GAS was calculated by computing counts per cell in the gene body and promoter region.

**scRNA-seq PCa.** The data consists of six biopsies from three patients, four radical prostatectomies, and four radical prostatectomies with matched normal samples sequenced using the Seq-Well S^3 protocol[33]. Analysis was run using the Snapshot 7 pipeline designed for drop-seq data. Cell types were generated using Seurat module scores calculated on sets of cell-type-specific signatures. Further clustering identified epithelial cells which were used in our study. For classification, we used malignant vs non-malignant cells which were identified by further analysis of the gene signatures.

**scRNA-seq NSCLC.** The data was accessed via figshare (https://figshare.com/collections/An_integrated_single-cell_transcriptomic_dataset_for1_non-small_cell_lung_cancer/6222221/3) and two cohorts were used (GSE136246 and GSE127465) that spanned across biopsies from multiple patients (Table in Fig. 6a). Both cohorts contain samples from patients diagnosed with LUAD and LUSC. Cell types were provided with the data and were transformed to be cancer vs. healthy cells. These were then used as truth labels to train and test scMKL. All samples from both cohorts were sequenced using Illumina's NextSeq platform, and output reads were processed using different software pipelines into count tables. Before applying scMKL, transcripts that are not in both datasets were removed.

### scMKL framework
To achieve interpretable multiomics data analysis, we developed scMKL, an MKL framework that integrates diverse feature representations (i.e., views) across various data modalities using different similarity measures or kernels (Fig. 1a, S6a). Unlike early fusion (concatenated features fed into a single learner) and late combination (combining classifiers trained on different features), scMKL performs intermediate fusion[54]. It merges multiple data sources at the kernel level before classification to capture complex cross-modal interactions without the high dimensionality challenges of *early fusion* or the potential loss of information in *late fusion*.

Kernel methods are widely used in biological applications[55,56]; however, scale poorly with large datasets like single-cell omics. To address this, we leveraged RFF[57] to approximate nonlinear kernels efficiently. This technique allows scMKL to retain the high accuracy of kernel methods while scaling to tens of thousands of cells. We paired this with GL as our base classifier, which selects biologically meaningful feature groups by enforcing group-wise sparsity[58].

**Multiple Kernel Learning.** A kernel function is a tool used in computer science for pattern separation by mapping data to a higher-dimensional space and calculating the inner product between them, without needing to know the exact mapping in the original space (i.e., functions that measure similarity between data points). We used MKL in two ways: first, to combine sets of features (e.g., different views of the data, such as gene sets) and second, to combine data modalities (e.g., omics such as RNA and ATAC). Instead of trying to find which pathways or omics work best, a learning method does the picking for us or may use a combination of them.

Consider a learning problem with input–omic data and output–cell phenotypes data $(\boldsymbol{x}_i, y_i)_{i=1}^{N}$ where $\boldsymbol{x}_i \in \boldsymbol{X}$ and $y_i \in \boldsymbol{Y}$ with $y_i \in \{-1, +1\}$ is its class label. There are several kernel functions successfully used in the

literature. In this study, we used Gaussian, Laplacian, and Cauchy kernels.

$$k_{GAU}\left(\boldsymbol{x}_i, \boldsymbol{x}_j\right) = \exp\left(-\frac{||(\boldsymbol{x}_i - \boldsymbol{x}_j)||_2^2}{s^2}\right)$$

$$k_{LAP}\left(\boldsymbol{x}_i, \boldsymbol{x}_j\right) = \exp\left(-||\boldsymbol{x}_i - \boldsymbol{x}_j||_1\right)$$

$$k_{CAU}\left(\boldsymbol{x}_i, \boldsymbol{x}_j\right) = \prod_d \frac{2}{1 + \left(\boldsymbol{x}_i - \boldsymbol{x}_j\right)_d^2}$$

where $s^2 \in \boldsymbol{R}$. Selecting the kernel function $k(\cdot, \cdot)$ and its parameters (e.g., $s$) is an important issue in training. We used a heuristic to calculate kernel widths (i.e., $s$) by the mean of the distance matrix, calculated on the training data using 2000 randomly selected cells and 200 features.

With MKL, we combine multiple kernels:

$$k_\eta\left(\boldsymbol{x}_i, \boldsymbol{x}_j\right) = f_\eta\left(k_m\left(\boldsymbol{x}_i^m, \boldsymbol{x}_j^m\right)_{m=1}^P\right)$$

where the combination function, $f_\eta : \boldsymbol{R}^P \to \boldsymbol{R}$, can be a linear or a non-linear function.

We calculated kernels with inputs from different representations, different sources, or modalities.

## Multiview learning (Integrating prior information)

Multiview refers to the technique where we approach problems from multiple perspectives to gain context from each. Each view was considered a different feature representation. Specifically, we can group biological features such as genes and ATAC peaks by previously identified patterns, such as function, pathway, or mechanism. Here, we use gene and ATAC peak sets defined to capture signals between individual features that drive differences in cell type, cell state, or treatment response.

**RNA-view.** Given $P_{RNA}$ gene sets $\left\{G_i^{RNA}\right\}_{i=1}^{P_{RNA}}$ we created kernels using RNA views for each gene set $G_i^{RNA} = \left\{\boldsymbol{x}^j\right\}_{j=1}^{|G_i^{RNA}|}$ all $\boldsymbol{x}^j \in G_i^{RNA}$ so that the MKL is formulated as follows:

$$k_\eta^{RNA}\left(\boldsymbol{x}_i, \boldsymbol{x}_j\right) = f_\eta^{RNA}\left(k_{G_m}^{RNA}\left(\boldsymbol{x}_i^{G_m^{RNA}}, \boldsymbol{x}_j^{G_m^{RNA}}\right)_{m=1}^{P_{RNA}}\right)$$

where the combination function learns the weights of each kernel $k_{G_m}^{RNA}$ via a classifier, in our case GL.

**ATAC-view.** Given input–omic data (ATAC) and output–cell phenotypes data $\left(\boldsymbol{x}_i, y_i\right)_{i=1}^N$ where $\boldsymbol{x}_i \in X^{ATAC}$. To create kernels for the ATAC-features–peak using gene sets, we first map peaks to the closest genes using a window size of 5 kb: $g_i = \{\boldsymbol{x}^r\}_{r=1}$ where $g_i \in G^{RNA}$ is a gene feature and $\boldsymbol{x}^s$ are the chromatin regions intersecting with $g_i$'s gene body region within 5 kb. Then, we calculate kernels using ATAC-features combining peaks according to their corresponding genes and the gene set info (See section Using gene set derived peak groupings). Given $P_{RNA}$ gene sets, we created kernels using ATAC features for each gene set:

$$k_\eta^{ATAC}\left(\boldsymbol{x}_i, \boldsymbol{x}_j\right) = f_\eta^{ATAC}\left(k_{G_m}^{ATAC}\left(\boldsymbol{x}_i^{G_m^{ATAC}}, \boldsymbol{x}_j^{G_m^{ATAC}}\right)_{m=1}^{P_{ATAC}}\right)$$

where $P_{ATAC} = P_{RNA}$ and $G_i^{RNA} = \{\boldsymbol{x}^j\}_{j=1}^{|G_i^{RNA}|} \Rightarrow$ $G_i^{ATAC} = \left\{\left(\{\boldsymbol{x}^r\}_{r=1}\right)^j\right\}_{j=1}^{|G_i^{RNA}|} = \{\boldsymbol{x}^r\}_{r=1}^{|G_i^{ATAC}|}$

When given $P_{ATAC}$ ATAC peak sets (e.g., bed files specific to TF factors), $\left\{G_i^{ATAC}\right\}_{i=1}^{P_{ATAC}}$ we used them as they are $G_i^{ATAC} = \{p^j\}_{j=1}^{|G_i^{ATAC}|}$ all $p^j \in G_i^{ATAC}$ and then the same MKL formulation above for ATAC follows.

## Multimodal learning (Integrating multiomics in a unified model)

scMKL enables multiomic integration in addition to incorporating prior information. We unlock interactions between data modalities by combining observations from transcriptional state, chromatin accessibility that may not be directly observable from an individual modality (referred to here as unimodal experiments). Given input–multiomic data (RNA + ATAC) $\left(\boldsymbol{x}_i, \boldsymbol{x}'_i, y_i\right)_{i=1}^N$ where $\boldsymbol{x}_i$ are the RNA features and $\boldsymbol{x}'_i$ are the ATAC features, MKL formulation is:

$$k_\eta^{RNA+ATAC}\left(\left(\boldsymbol{x}_i, \boldsymbol{x}'_{i,}\right), \left(\boldsymbol{x}_j, \boldsymbol{x}'_{j,}\right)\right)$$
$$= f_\eta^{RNA+ATAC}\left(k_{G_m^{RNA}}\left(\boldsymbol{x}_i^{G_m^{RNA}}, \boldsymbol{x}_j^{G_m^{RNA}}\right)_{m=1}^{P_{RNA}} + k_{G_m^{ATAC}}\left(\boldsymbol{x}'_i{}^{G_m^{ATAC}}, \boldsymbol{x}'_j{}^{G_m^{ATAC}}\right)_{m=1}^{P_{ATAC}}\right).$$

**Random Fourier Features.** Random features are a technique used to accelerate supervised learning algorithms, allowing for scalable processing of large datasets. This approach enables accurate learning algorithms that operate at high speeds and are easy to implement.

**Calculating approximate kernels.** For each feature grouping $G_i$, we computed RFF by multiplying each input matrix by a random vector $\boldsymbol{\omega} \in \boldsymbol{R}^{|G_i|}$ sampled from the Fourier transform of the kernel function $p(\omega)$. Finally, we applied sine and cosine functions. The resulting $\boldsymbol{Z}_i \in \boldsymbol{R}^{N \times 2D}$ matrices for each grouping were passed to GL as input:

$$\boldsymbol{Z}_i = \sqrt{\frac{1}{D}}\left[\cos\left(\boldsymbol{w}_1^T \boldsymbol{X}\right) \ldots \cos\left(\boldsymbol{w}_D^T \boldsymbol{X}\right) \sin\left(\boldsymbol{w}_1^T \boldsymbol{X}\right) \ldots \sin\left(\boldsymbol{w}_D^T \boldsymbol{X}\right)\right]$$

where $\boldsymbol{X} \in \boldsymbol{R}^{N \times |G_i|}$ and the columns of the matrix $\boldsymbol{X}$, $\boldsymbol{x}^j \in G_i$, $D$ ($D < N$) is the number of RFF.

For RNA and TF-IDF transformed data, we use a Gaussian Kernel function, as it is well studied and proven to be effective for these data types[56,58].

For ATAC data, we tested the mapping $\boldsymbol{Z}$ calculated for Gaussian, Laplacian, and Cauchy kernel functions and showed that the Laplacian kernel results in sparser solutions, with higher AUROC (Fig. S9a).

For RNA data, $\boldsymbol{Z}_i$ was calculated using $\boldsymbol{X} \in X_{RNA}$ and the columns of the matrix $\boldsymbol{X}$, $\boldsymbol{x}^i \in G_i^{RNA}$ while for ATAC, $\boldsymbol{X} \in X_{ATAC}$ and the columns of the matrix $\boldsymbol{X}$, $\boldsymbol{x}^i \in G_i^{ATAC}$. For RNA + ATAC we concatenate the $\boldsymbol{Z}$'s calculated for RNA and ATAC for joint optimization in GL.

Previous studies have demonstrated that accurate approximation of a kernel function requires $D \geq \sqrt{N} \times \log\left(\log(N)\right)$ for $N$ samples[59].

We showed that using higher $D$ values results in higher AUROC scores, which reached saturation around the $D$ value calculated using the equation above (Fig. S9b). We used the above equation as the default $D$ value for most experiments; the only exceptions are experiments using JASPAR feature groupings. For JASPAR experiments, we decreased $D$–100 to accommodate the large number of feature groupings.

**Group Lasso.** GL is a regularization technique in machine learning aimed at inducing sparsity at the group level. Unlike Lasso regularization, which penalizes individual features independently, GL operates on predefined groups of features. These groups can be determined based on domain knowledge or feature similarities.

The objective of GL is to encourage the selection of entire groups of features while simultaneously shrinking the coefficients of less relevant groups towards zero. The model achieves group-level sparsity by adding a

penalty term to the loss function, which is the sum of the L2-norms of the coefficients within each group.

$$minimize\left(\frac{1}{2} \times ||y - \mathbf{Z}\boldsymbol{\beta}||^2 + \lambda \times \sum ||\boldsymbol{\beta}_g||_2\right)$$

where $\mathbf{Z}$ is the mapping of the kernels $\boldsymbol{\beta}$ is the vector of coefficients, $\boldsymbol{\beta}_g$ represents the coefficients within each group g (i.e., $G^{RNA}$, $G^{ATAC}$, $G^{RNA+ATAC}$), $\lambda$ is the regularization parameter controlling the strength of the penalty (i.e., solution sparsity/number of groupings to be weighted).

Each $\mathbf{Z}$ calculated on RNA and ATAC groupings passed to GL with the information that its columns weighted as a group. We optimized $\lambda$ using four-fold cross-validation. We determined $\lambda$ to evaluate by identifying the $\lambda$'s that select one group and all groups and calculating the eight values linearly spaced between the two. This selection process was repeated for each experiment.

In the multiview setting, scMKL constructs modality-specific kernels using expert-curated biological groupings: pathway-based gene groups for RNA-seq and TFBS-associated regions for ATAC-seq. Each kernel is approximated using RFF ($Z_i$) and assigned a learned weight ($\eta_i$), quantifying the importance of each group. Interpretability is achieved by visualizing feature group selection frequency and kernel weights using dot plots, enabling biological insight into which pathways or TFs drive classification.

## Experimental settings and performance metrics
We created three scenarios to perform experiments for scRNA-seq, scATAC-seq, and sc-multiomics.

**TF-IDF transformation**. Term frequency inverse document frequency (TF-IDF) normalization is a technique from natural language processing to find the relative importance of words in documents:

$$\mathbf{X} \times \log\left(\frac{(1 + C)}{1 + \Sigma_{j=1}^{C}(\boldsymbol{x}^j > 0)} + 1\right) \text{ for } \mathbf{X} \in \mathbf{R}^{N \times C}.$$

**TF groupings**. For ATAC-specific feature groupings, we use motifs and TFs from the JASPAR and Cistrome Databases. To incorporate information from JASPAR, we performed a motif enrichment analysis for each ATAC dataset to assign peaks to motifs[13]. For Cistrome groupings, we used tissue-specific groupings from published literature[12]. In the case of multiple studies publishing data on the same TF, we used the study that annotated the most peaks in their final data and removed any with fewer than 500 peaks. We applied these groupings to our datasets by looking for overlap in the genomic region between our binned peaks and the published peaks.

## Implementation of scMKL
We showed the efficacy of MKL for bulk and scRNA-seq data for phenotype and survival predictions[58]. Our framework implements a similar approach for scATAC-seq and sc-multiomics. Our Python implementation improved scalability and accuracy compared to the R implementation. scMKL also takes advantage of the AnnData framework for easy integration into the existing sc-verse[60].

**Interpretation dotplots**. For each experiment, we averaged kernel selection frequencies and weights across $\lambda$ values to obtain a more robust view of feature group importance. We retained the top 8 RNA and top 8 ATAC feature groups ranked by selection frequency. To normalize across solutions with varying sparsity, we applied min-max scaling separately to kernel weights and selection frequencies, within each modality (RNA, ATAC, and RNA + ATAC). In visualizations, color denotes modality, size indicates selection frequency, and opacity reflects scaled kernel weight.

## Baseline algorithms
**Supervised algorithms**. In this study, we compared our method against four algorithms, namely, SVM, EasyMKL, XGBoost, and MLP.

XGBoost is an ensemble tree-based classification algorithm previously noted for its scalability and improved performance relative to random forest classifiers[61]. We used the scikit-learn implementation of XGBoost with default parameters and performed 4-fold cross-validation to optimize the max depth parameter in each replication[62].

MLP is a class of feed-forward neural networks with fully connected layers. Here, we use a 5-hidden-layer architecture with 512 nodes, each using a ReLU activation function and separated by a dropout layer with 10% dropout. The final layer is a sigmoid activation function for binary classification. We implement our MLP in Keras and perform 4-fold cross-validation to tune hyperparameters, including batch size[63].

EasyMKL is a recent MKL algorithm implemented in the MKLpy package. It is designed for scalability and scales linearly in time and memory with the number of kernels. We used the method with default parameters and performed 4-fold cross-validation to optimize the regularization parameter $\lambda$ over the set [0, 0.25, 0.5, 0.75, 1]. We experimented with using 50 hallmark kernels for RNA.

SVMs are a class of kernel-based classifiers previously shown to be effective for disease classification using bulk RNA data. Here, we employed the SVC model from *scikit-learn* with default settings, including the radial basis function kernel. We performed 4-fold cross-validation over the parameter C, tuning across the set [0.01, 0.1, 1, 10, 100].

**Feature selection**. For comparison methods, we applied three different feature selection or representation strategies per modality when computationally feasible. For RNA, we used:
1. All genes
2. Hallmark genes
3. Principal Component Analysis (PCA)

For ATAC, we used:
1. Most variable peaks
2. Hallmark peaks
3. Latent Semantic Indexing (LSI)

PCA and LSI were implemented using the PCA and TruncatedSVD modules from *scikit-learn*. For PCA, we log-normalized and z-scored raw RNA counts, fitting the transformation on the training data and applying it to the test set. For LSI, we first computed TF-IDF normalization. We subsequently log-normalized and z-scored the TF-IDF matrix. For PCA, we retained the top 50 components. For LSI, we kept the top 50 singular values and excluded the first component, which is often associated with sequencing depth.

## Unsupervised algorithms
**Standard single-cell analysis workflow**. We followed a standard single-cell analysis using scanpy (v1.10.1)[27]. For scRNA assays, we removed genes expressed in fewer than three cells and cells that expressed fewer than 100 genes. Samples with more than 30% mitochondrial gene expression were also removed. After log transformation and standard scaling, scanpy's (v1.4.2) built-in rank_genes_group function identified DEGs using a Wilcoxon rank-sum test. Using a Benjamini-Hochberg adjusted *p*-value significance cutoff of 0.05, we identified DEGs for MCF-7.

Once DEGs were calculated with scanpy, we input the results into GSEApy's (v1.1.3) enrichr and prerank functions to calculate gene set enrichment using Hallmark genes. Enrichr applies Fisher's exact test with a combined score based on the log-transformed *p*-value and z-score, while GSEA Preranked calculates enrichment using a Kolmogorov–Smirnov-like statistic and evaluates significance via permutation testing. GSEA yielded adjusted *p*-values for each gene set in the Hallmark library, allowing a fair comparison of top gene sets identified by GSEA and scMKL. We used sklearn.metric's silhouette_samples function to calculate silhouette scores.

## Resource usage

All experiments were performed on an Intel(R) Xeon(R) Gold 6326 CPU @ 2.60 GHz (x86_64 architecture) with M386A8K40CM@-CTD Samsung RAM module. We aimed to complete all experiments across comparison methods; however, we omitted bootstrapped runs that required over 50 GB of RAM or exceeded 12 hours of runtime. These included: MLP with all genes, LSI with scATAC-PCa, XGBoost, and SVM on LUAD and LUSC datasets.

## Randomized and noisy pathway experiments

To assess the robustness of pathway-driven grouping, we performed two perturbation experiments:

1. **Corrupted pathways:** A proportion of features in each group was replaced with randomly selected features outside the group. This disrupts the group structure and introduces irrelavant features to the group. We evaluated corruption levels of 0, 20, 40, 60, 80, and 100%.
2. **Noisy pathways:** Features in the biological groups were preserved, but we appended randomly permuted values of existing features to each group (i.e., uncorrelated with the outcome). This maintains the data distribution but adds uninformative noise features. Noise was added at levels of 0, 50, 100, 200, 400, and 800% relative to the original group size.

## Multimodal interpretation ablation experiment

To evaluate redundancy and interaction in multimodal feature groups, we performed targeted ablations of estrogen-related groups in the RNA Hallmark + ATAC JASPAR MCF7 experiment. The following groups were individually or jointly removed: Hallmark ER Early, Hallmark ER Late, ESR1, ESR2, Hallmark ER Early + ER Late, Hallmark ER Early + ESR1.

We evaluated performance changes and group selection post-exclusion of this feature groups. To quantify impact, we generated volcano plots comparing kernel weight differences (x-axis) with the $-\log10(p\ value)$ from a Wilcoxon rank-sum test on kernel norms (y-axis) between baseline and ablated models.

## Statistics and Reproducibility

We compared the prediction performance of scMKL vs. MLP, XGBoost, and SVM using a one-sided Wilcoxon ranked-sign test. Unless otherwise noted, these results were calculated on 100 randomly stratified train-test splits. The Benjamini-Hochberg correction was done using the multipletests function from the statsmodels Python package.

## Reporting summary

Further information on research design is available in the Nature Portfolio Reporting Summary linked to this article.

## Data availability

No new patient data have been generated in the study. The multiome data on breast cancer cell lines is available under the accession number GSE154873[53]. Source data files for SLL multiome data are available on the 10x genomics website as 'Flash-Frozen Lymph Node with B Cell Lymphoma at https://www.10xgenomics.com/resources/datasets/. The raw single-cell ATAC-sequencing files and processed data files are available under the GSE accession number: GSE171559[32]. Raw single-cell RNA-sequencing FASTQ files and gene expression matrices files are available in the GSE accession number: GSE176031[33]. The raw NSCLC data can be found using accession numbers GSE136246 and GSE127465[40,41]. Additionally, the Seurat object with cell annotations for both datasets can be found on figshare (https://figshare.com/collections/_/6222221)[64].

## Code availability

An open-source Python implementation of the scMKL is accessible at https://github.com/ohsu-cedar-comp-hub/scMKL. The scMKL distribution is also available on the Python Package Index (https://pypi.org/project/scmkl/). Results were generated using version 0.1.6 of scMKL. The shiny for python app for visualizing all the results in more detail in this paper are available at https://huggingface.co/spaces/scMKL-team/scMKL_analysis.

Code for comparison algorithms and scalability tests are accessible at https://github.com/ohsu-cedar-comp-hub/scMKL_paper. Results generated by scMKL experiments, with the scripts used to generate them are accessible at https://doi.org/10.5281/zenodo.15397923[65].

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

## Acknowledgements

The research reported in this publication used computational infrastructure supported by the Office of Research Infrastructure Programs, Office Of The Director, of the National Institutes of Health under Award Number S10OD034224. The content is solely the responsibility of the authors and does not necessarily represent the official views of the National Institutes of Health. Figure 1 was created with BioRender.com. We thank Lindsey Cauthen for her feedback on the manuscript and her revisions. We thank Hisham Mohammed for his input on utilizing the Cistrome database for analyzing breast cancer data and James McGann for his support as the project facilitator. This project was supported by funding from the Cancer Early Detection Advanced Research Center (CEDAR) (CEDAR Full 2022-1492) at Oregon Health & Science University, Knight Cancer Institute (C.A.).

## Author contributions

C.A., M.G., and S.E. conceived of the project and developed the theoretical foundations of the method. C.A., S.K., and I.V. implemented scMKL. C.A., S.K., and I.V. designed and performed the computational data analysis. S.E.E. interpreted the PCa results. I.V. ran the comparisons with standard single-cell workflows, created the shiny app, and assembled the scMKL pipeline into a Python package. C.A., S.K., I.V., and S.E.E. wrote the manuscript with input from all authors. C.A. supervised all aspects of the project.

## Competing interests

The authors declare no competing interests.

## Additional information

Primary Handling Editors: Kuangyu Yen and Aylin Bircan, Kaliya Georgieva. A peer review file is available.

