## [Transparent Peer Review file · Communications Biology]

Interpretable and Integrative Analysis of Single-Cell Multiomics with scMKL

Corresponding Author: Dr Cigdem Ak

Version 0:

Reviewer comments:

Reviewer #1

(Remarks to the Author)

The manuscript presents an innovative approach, scMKL, for interpretable and integrative single-cell multiomics analysis. By leveraging multiple kernel learning, the method combines predictive accuracy with interpretability, addressing key challenges in single-cell data analysis. Its application across various datasets and cancer types underscores its versatility and potential for significant contributions to the field of computational biology. I am appreciating authors effort in this work. However I have some major and minor comments.

Major Comments:

1. The development of scMKL demonstrates a meaningful advance by integrating multiple data modalities while maintaining interpretability. The manuscript could benefit from explicitly comparing scMKL's novelty over other existing kernel-based methods in more detail.
2. While AUROC and F1 scores were used to demonstrate scMKL's superiority, the manuscript should also discuss other metrics (e.g., precision-recall curves) for imbalanced datasets, if applicable.
3. The successful application of transfer learning between datasets (e.g., MCF-7 and T-47D cell lines) is a strong point. Expanding on the robustness of this transferability, especially under varying preprocessing pipelines, would strengthen the manuscript.
4. While scMKL's interpretability is a central claim, the manuscript could include more detailed examples of actionable insights derived from interpretability. For instance, showing how selected pathways informed experimental validation.
5. Although scalability metrics are included, discussing computational requirements and reproducibility tools (e.g., code repository availability) would add value to the reader.

Minor Comments:

1. Ensure consistency in terminology, e.g., "multiomics" vs. "multi-omics."
2. Avoid redundancy, such as in repeated explanations of kernel functions.
3. Enhance figure captions for clarity; provide brief methods or results within captions (e.g., for Figure 2 and 5).
4. Include a detailed legend for Table 1 (dataset information).
5. Expand citations for comparison methods and prior information databases to ensure comprehensive coverage.
6. Specify which pathway enrichment tests were used in benchmarking scMKL against standard workflows (e.g., GSEA).
7. Define all acronyms at first use and include a table for quick reference (e.g., AUROC, TF-IDF).

The manuscript is an impressive contribution to the field, offering a well-rounded, interpretable solution for analyzing complex single-cell data. With enhancements to clarity, expanded reproducibility discussions, and deeper examples of interpretability, this work will be a strong candidate for publication in computational biology or systems biology journals.

Reviewer #2

(Remarks to the Author)

This manuscript presents a method based on scMKL, designed for integrative single-cell multiomics analysis using RNA

and ATAC data. The authors demonstrate the potential of this method in improving classification accuracy and interpreting biological phenomena. The study offers significant innovation and methodological advancement. However, while the paper achieves notable results in multi-modal data integration and model application, there remain several issues that need addressing, particularly regarding interpretability, computational efficiency, generalization ability, and comparisons of preprocessing strategies. Below are specific reviewer concerns.

1. Interpretability of Multi-Modal Data Integration and Model Transparency

Concern: The paper emphasizes the advantages of scMKL in integrating multi-modal data (RNA + ATAC), but the interpretability of the integration process remains somewhat unclear. While the authors demonstrate improved classification accuracy, they provide limited explanation regarding how the model handles the interaction between different modalities. Specifically, how the model avoids information loss or redundancy between modalities and ensures integration does not lead to overfitting is not sufficiently discussed.

Recommendation: The authors should elaborate on the specific strategies used in the scMKL model for handling multi-modal data, especially on how biological information is effectively captured during integration. Additionally, a more detailed discussion on model transparency and interpretability is needed, such as explaining the decision-making process, feature importance, etc. Further visualization techniques (e.g., heatmaps, interpretable model diagrams) would help readers better understand how different data modalities influence the final decision of the model.

2. Computational Efficiency of ATAC Feature Selection

Concern: The paper mentions that the high-dimensionality of ATAC data limits computational resources, and thus only the most variable ATAC features are used for analysis. However, the process of ATAC feature selection could affect the model's overall performance, particularly under resource constraints. While the authors claim that scMKL outperforms traditional classifiers in these conditions, the balance between computational resource constraints and classification accuracy is not thoroughly explored.

Recommendation: The authors should consider optimizing the ATAC feature selection process, potentially by introducing more efficient feature selection strategies or dimensionality reduction methods. Techniques like PCA, t-SNE, or sparse feature selection methods (e.g., LASSO) could help improve computational efficiency while retaining sufficient biological information. Additionally, comparative experiments assessing the impact of different feature selection strategies under resource-limited conditions would be beneficial.

3. Model Generalization and Cross-Dataset Validation

Concern: Although the authors demonstrate scMKL's superior performance across different cell lines and datasets, more validation is needed to assess its generalization ability across datasets. Specifically, it remains unclear whether the model can maintain high performance across different tumor types and cell types, such as comparing data from prostate cancer versus breast cancer cell lines.

Recommendation: The authors should further validate scMKL's generalization capability, particularly on a broader range of cancer datasets. Testing the model on different tumor types and clinical backgrounds would help assess its robustness and performance consistency across datasets. Exploring ways to optimize the model to handle diverse tumor types would further enhance its clinical applicability.

4. Comparison and Impact of Different Preprocessing Strategies

Concern: The manuscript compares scMKL with standard single-cell analysis workflows (such as Seurat and scanpy) and highlights the impact of different preprocessing strategies on model performance. However, preprocessing steps can significantly influence the final results, and there is insufficient discussion regarding how the choice of genes, dimensionality reduction, and differential analysis steps affects the outcome.

Recommendation: A more in-depth discussion of how different preprocessing steps (e.g., gene selection, dimensionality reduction, data normalization) influence the performance of the scMKL model is needed. Additionally, providing more experimental data comparing different preprocessing strategies and their effects on model performance would help readers better understand the importance of preprocessing choices in this context.

5. Minor Formatting Issues

Concern: There are some formatting inconsistencies in the paper, such as differences in font between Table 1 and Table 2. Additionally, in Figure 2 and other figures, boxplots lack statistical significance annotations, which are essential for guiding the reader on whether the observed differences between methods are statistically significant.

Recommendation: The authors should standardize the font across tables and figures for consistency. Furthermore, statistical significance annotations should be included in the boxplots in Figure 2 and other relevant figures to improve clarity and help readers assess the statistical validity of the reported differences.

In conclusion, while the paper presents an innovative single-cell multiomics analysis method and demonstrates its strengths in RNA + ATAC data integration, there are several areas that need further improvement, particularly regarding interpretability, computational efficiency, generalization, and preprocessing strategy comparisons. The authors are encouraged to address these concerns in greater detail and provide additional experimental data to strengthen the scientific rigor and application potential of their approach.

Version 1:

Reviewer comments:

Reviewer #2

(Remarks to the Author)

I am satisfied with the authors' thorough and thoughtful responses to my previous comments.

Note: Reviewer comments are in blue, our responses are in black, and text added to the revised manuscript is italicized.

Reviewers' comments:

Reviewer #1 (Remarks to the Author):

The manuscript presents an innovative approach, scMKL, for interpretable and integrative single-cell multiomics analysis. By leveraging multiple kernel learning, the method combines predictive accuracy with interpretability, addressing key challenges in single-cell data analysis. Its application across various datasets and cancer types underscores its versatility and potential for significant contributions to the field of computational biology. I am appreciating authors effort in this work. However I have some major and minor comments.

Major Comments:

1. The development of scMKL demonstrates a meaningful advance by integrating multiple data modalities while maintaining interpretability. The manuscript could benefit from explicitly comparing scMKL's novelty over other existing kernel-based methods in more detail.

We appreciate the reviewer's recognition of scMKL as a meaningful advance in integrating multiple data modalities while maintaining interpretability. To clarify scMKL's novelty over existing kernel-based methods, we have revised the manuscript to explicitly compare its, scalability, interpretability, and applicability to single-cell multiomics, including empirical comparisons with both SVM (with RBF kernel) and EasyMKL. Our results demonstrate that scMKL achieves comparable or superior classification performance while also offering improved computational efficiency in terms of runtime and memory usage. These findings are now clearly presented in the Results and Discussion sections.

We added the following discussion to the Section "scMKL outperforms state-of-the-art classification algorithms" and updated computational efficiency figures as shown below:

Lines 294-327: "scMKL is the first multiple kernel learning (MKL) framework to integrate RNA and ATAC data at the single-cell level, overcoming key scalability and interpretability limitations of traditional kernel-based approaches. While several MKL methods have been applied to bulk multiomics datasets, such as TCGA (e.g., Wilson et al., 2019; Briscik et al., 2024), they operate at the patient level and are not scalable to the high dimensionality and high cell counts of single-cell data. Moreover, they lack interpretability at the feature or modality level.

A few kernel-based methods have been developed for single-cell RNA-seq analysis, including SIMLR (Wang et al., 2017) and the spectral clustering approach of Ren Qi et al. (2021), but these are limited to small datasets, do not support multimodal integration, and offer limited biological interpretability. Other patient-level approaches, such as PIMKL (Manica et al., 2019), employ pathway-induced kernels for bulk RNA data but are not scalable to single-cell resolution and do not incorporate epigenomic modalities like ATAC.

*scMKL addresses these gaps through innovations in both computational efficiency and biological interpretability. It employs RFF to efficiently approximate nonlinear kernels, reducing computational complexity from $O(N^2)$ to $O(N)$, and uses Group Lasso regularization for sparse, modality-aware feature selection. This design enables *scMKL* to scale to large, high-dimensional single-cell datasets while maintaining interpretability by identifying the contributions of specific features and modalities.*

*In contrast to prior methods, *scMKL* uses omic-specific kernels tailored to each modality’s data distribution, integrates prior knowledge from both RNA and ATAC domains, and provides interpretable weights reflecting the importance of features within and across modalities—offering direct insight into the biological signals driving classification.*

*To further demonstrate *scMKL*’s advantages, we first benchmarked it against standard kernel-based classifiers, including SVM and EasyMKL. SVM is restricted to a single kernel, while EasyMKL is limited in scalability. In comparisons on our smallest dataset, *scMKL* achieved superior classification accuracy while training $7\times$ faster and using $12\times$ less memory than EasyMKL (Fig 2d). Although we matched the number of pathway-informed kernels for fairness, existing methods became computationally impractical at larger scales. These results, detailed in Figures 2c–d and Figure 2a-b, highlight *scMKL*’s ability to scale multimodal integration without compromising performance or interpretability.”*

Figure 2c-d:

Figure 2. *scMKL* provides scalable, flexible, and accurate predictions across diverse datasets **c.** Comparison of scalability cost: Analysis of training time in seconds and memory in GB across different sample sizes (1k-6k cells) for the four machine learning models (MLP, XGBoost, SVM and *scMKL*) using all genes, Hallmark genes, Hallmark peaks, and most variable peaks. **d.** Comparison of MKL methods (*EasyMKL* vs. *scMKL*), showing AUROC, memory usage, and runtime as a function of sample size.

Figure S2a:

Figure S2. scMKL provides scalable, flexible, and accurate predictions across diverse datasets and identifies biologically relevant mechanisms through pathway/TF selection
a. Training memory usage comparison: Analysis of training memory (in GB) and time (in seconds) usage across the four machine learning models. *, **, *** are used to denote statistical significance from Wilcoxon test p values < 0.05, 0.01, and 0.001 respectively. Gold indicates scMKL had significantly better performance; black indicates the benchmark algorithm had significantly higher performance.

2. While AUROC and F1 scores were used to demonstrate scMKL's superiority, the manuscript should also discuss other metrics (e.g., precision-recall curves) for imbalanced datasets, if applicable.

We thank the reviewer for this helpful suggestion. We further evaluated scMKL using additional metrics, including precision-recall curves, to assess performance under class imbalance (see below Figure S1-for RNA and ATAC predictions- and 2a -RNA+ATAC predictions-). While scMKL performed comparably to other methods overall, results varied with class distribution, indicating sensitivity to imbalance—a common issue among classifiers. This can be addressed using subsampling or other balancing techniques.

To explore this, we analyzed a non-small-cell lung cancer (NSCLC) data with varying cancer-to-healthy cell ratios. Given the limited number of cancer cells, healthy cells were randomly subsampled at different ratios to create balanced training cohorts. For each ratio, ten replicates were generated using different healthy cell subsets. Our results show that balancing classes during training improves prediction accuracy (see below **Figure 6b**).

Notably, we also observed that multimodal integration enhances scMKL's robustness and performance-especially dramatically in F1, precision and recall, even in imbalanced settings (See below **Fig S4a-b**).

We added the following texts in the manuscript:

Lines 96-99: “To evaluate generalizability further, we systematically benchmarked scMKL on challenging classification tasks to assess its generalizability across tumor subtypes, its robustness to class imbalance, and its ability to transfer across independent non-small cell lung cancer scRNA-seq datasets under biologically and technically distinct conditions.”

Lines 355-362: “We also reported F1, precision, and recall scores as additional measures of predictive accuracy (Fig S1a-b). Across all datasets, scMKL in most cases outperforms or matches the best baseline with a smaller variance in AUROC. In F1, precision, and recall, for the two highly class-imbalanced datasets—LUAD and LUSC (class ratios of 12:1 and 7:1, respectively) scMKL performed poorly, which we investigate in the following sections (Fig 6). Due to the large sample sizes of LUAD and LUSC datasets, it was not feasible to run all state-of-the-art methods without exceeding memory limits. To evaluate computational efficiency, we benchmarked runtime and memory usage with upper bounds of 12h and 50 GB (Fig 2c-d and S2a-b).”

*Lines 529-533: “In **Fig S2c**, we showed that JAPAR and Hallmark binary peak groups have overall higher F1, precision, and recall scores. With Hallmark+JASPAR multimodal integration, scMKL achieved statistically significant improvements in predictive performance across all datasets and evaluation metrics, consistently outperforming all competing algorithms when using ATAC data (**Fig S4**).”*

Lines 1081-1085: In section “scMKL generalizes across cohorts and tumor subtypes while handling class

***Robustness to class imbalance:** To assess the effect of class imbalance, we varied the healthy-to-cancer ratio in training and evaluated classification performance across ten replicates per condition (Fig 6b). scMKL remained stable across a wide imbalance range (Fig 6b, S7a). Performance improved with greater feature group inclusion (lower regularization) under skewed ratios, while moderate sparsity (higher λ) yielded optimal results near balance (0.6-1.6).”*

In addition, we updated the following figures with F1, Precision, and Recall:

a

Figure 3. scMKL enables accurate, interpretable multimodal classification and identifies biologically relevant mechanisms through robust feature selection.

a. Comparison of classification performance (AUROC, F1-score, precision, recall) between RNA (red), ATAC (blue), and RNA+ATAC (purple) models across MCF-7, T-47D, and SLL multimodal datasets. *, **, and *** denote significance determined by Wilcoxon tests for p-values < 0.05, 0.01, and 0.001 respectively. The color indicates which experiment had significantly higher performance.

Figure 5. scMKL stratifies cells from low- and high-grade prostate cancer tumors identifying factors associated with disease progression and subtypes.

a-b. Classification performance (AUROC, F1-score, precision, recall) between GAS (red), ATAC (blue), and combined GAS+ATAC (purple) models in PCa sc-ATAC-seq data. along with Hallmark pathway selection frequency and kernel importance scores, highlighting consistently selected biological pathways. . *, **, and *** denote significance determined by Wilcoxon tests for p-values < 0.05, 0.01, and 0.001 respectively. The color indicates which experiment had significantly higher performance.

Figure 6b. Impact of class imbalance on classification performance across metrics (AUROC, G-mean, F1-score, recall, and precision) under varying healthy-to-cancer cell ratios and regularization strengths (λ).

Figure S1. scMKL provides scalable, flexible, and accurate predictions across diverse datasets

a-b. Classification performance comparison: scMKL, MLP, XGBoost, and SVM were evaluated across seven datasets, including three multimodal datasets—MCF-7, T-47D, and SLL (panel b) and three unimodal datasets—PCa, LUAD, and LUSC scRNA-seq and PCa scATAC-seq (panel a). Darker color shades represent models trained on all features, while lighter shades denote models using Hallmark or MVF features for RNA and ATAC.

C

Figure S2. scMKL provides scalable, flexible, and accurate predictions across diverse datasets and identifies biologically relevant mechanisms through pathway/TF selection
c. Comparison of ATAC data groupings and transformations: Predictive performance for binary and TF-IDF normalized ATAC peaks using Hallmark gene sets, Cistrome and JASPAR TFBS as prior information. *, **, and *** denote significance determined by Wilcoxon tests for p-values < 0.05, 0.01, and 0.001 indicating which ATAC data transformation had significantly improved performance.

Figure S4. Biologically informed multimodal integration outperforms single-modality and state-of-the-art methods across diverse datasets a. Multimodal integration of RNA and ATAC

using Hallmark gene set-informed gene groups and JASPAR TFBS-informed peak groups consistently outperforms state-of-the-art algorithms across all datasets. *, **, *** are used to denote statistical significance from Wilcoxon test p values < 0.05, 0.01, and 0.001 respectively. Gold indicates scMKL had significantly better performance; black indicates the benchmark algorithm had significantly higher performance. **b.** Multimodal integration of RNA and ATAC using JASPAR-informed features not only surpasses single-modality predictions but also outperforms Hallmark-based multimodal integration across all datasets and evaluation metrics.

a

c

Figure S5. Hallmark pathways and Cistrome TFs in the RNA+ATAC multimodal setting.
a. *Hallmark pathways as prior information.* Comparison of classification performance (AUROC, F1-score, precision, recall) between RNA (red), ATAC (blue), and RNA+ATAC (purple) models

across MCF-7, T-47D, and SLL multimodal datasets. *, **, and *** denote significance determined by Wilcoxon tests for p-values < 0.05, 0.01, and 0.001 indicating which experiments resulted had significantly improved performance. **c. Hallmark pathways and Cistrome TF as prior information.** Comparison of classification performance (AUROC, F1-score, precision, recall) between RNA (red), ATAC (blue), and RNA+ATAC (purple) models across MCF-7 and T-47D multimodal datasets. . *, **, and *** denote significance determined by Wilcoxon tests for p-values < 0.05, 0.01, and 0.001 respectively. The color indicates which experiment had significantly higher performance.

3. The successful application of transfer learning between datasets (e.g., MCF-7 and T-47D cell lines) is a strong point. Expanding on the robustness of this transferability, especially under varying preprocessing pipelines, would strengthen the manuscript.

We evaluated scMKL's performance across datasets using multiple preprocessing pipelines, including raw counts, log normalization, TF-IDF, LSI, and the standard single-cell processing workflow implemented in Scanpy. scMKL remained highly comparable across all pipelines, demonstrating its robustness to variations in data processing. This consistency highlights scMKL's adaptability and suggests that it can be effectively applied across different preprocessing strategies without significant loss of performance (See below Figure S9).

To further assess the robustness of transferability, we conducted six distinct transfer learning experiments using two **independent NSCLC datasets** generated by **collected at different times and processed with different computational pipelines** (Trained on GSE136246, Maroni et al. (2021) and evaluated on GSE127465, Zilionis et al. (2019); See below Figure 6). These cases span multiple disease contexts and cohort compositions:

Case 1 trained on all cancer and a sampled subset of healthy cells, predicting on the full test cohort.

Case 2 used LUAD cancer and LUAD healthy cells in both training and testing.

Case 3 focused exclusively on LUSC patients for both training and testing.

Case 4 trained and tested on LUAD vs. LUSC cancer cells.

Case 5 trained on LUSC samples and tested on LUAD.

Case 6 trained on LUAD and tested on LUSC.

The train and test datasets originate from different studies that were collected at different times and processed using different software. Despite these differences, scMKL achieved strong and consistent classification performance across all six scenarios, reinforcing its robustness to cross-study variability and its capacity for generalization across datasets and tumor subtypes. We discuss these results in the section titled "scMKL generalizes across cohorts and tumor subtypes while handling class imbalance" between the lines 1064-1143.

Lines 469-505: "To evaluate the impact of different preprocessing strategies on scMKL, we conducted a series of experiments focused on gene selection, data normalization, and dimensionality reduction. scMKL does not rely on PCA or LSI prior to modeling. Instead, it leverages biologically informed groupings (e.g., pathways/gene sets or TF-informed peak groups), allowing for structured feature construction and selection. However, we benchmarked several preprocessing alternatives for both RNA and ATAC after construction of each feature group. For RNA, we compared PCA, log-normalization, raw counts, and standard scRNA-seq

preprocessing workflow (i.e., using scanpy, including filtering low-quality cells and genes, normalization, dimensionality reduction). For ATAC, we compared TF-IDF, LSI, binary counts, standard scATAC-seq preprocessing workflow (i.e., using scanpy/muon suite¹, including filtering low-quality cells and peaks, normalization, dimensionality reduction). scMKL demonstrated robust performance across a range of preprocessing methods, with metrics AUROC, F1-Score, Precision, and Recall consistently achieving high values (e.g., AUROC spanning from 0.9731 to 0.9999). Notably, the optimal preprocessing method varied depending on both the dataset and the specific performance metric, highlighting that while scMKL was consistently effective across different approaches, no single preprocessing technique proved universally superior. This variability underscores the adaptability of scMKL, ensuring reliable outcomes across diverse biological datasets and preprocessing strategies. Furthermore, filtering cells effectively reduces ambiguity, thereby enhancing data clarity between distinct classes. In this context, after standard processing workflows, the Binary method performed best across T47D, Lymphoma, and Prostate, while LSI outperformed the other methods in MCF7 (Fig S3).”

Figure S3. Comparison of preprocessing strategies for RNA and ATAC feature representations across multiple datasets **a.** RNA modality: AUROC, F1-score, precision, and recall for scMKL trained on RNA data from MCF7, T47D, SLL, and prostate cancer datasets. Comparisons include: PCA on log-normalized and z-scored RNA counts (50 components), standard workflow using all genes, and log(z)-transformed counts without dimensionality reduction. **b.** ATAC modality: AUROC, F1-score, precision, and recall for classifiers using ATAC peak counts from the same datasets. Comparisons include binary peak presence/absence

matrix, TF-IDF transformation, and LSI via TruncatedSVD (50 components, first component removed), as well as the standard workflow using most variable peaks.

a

Dataset	Sample #	QC-passed cell #	Stage	Gender	NSCLC subtype	Use of dataset
Maroni et al., 2021	24	53,190	I-IV	F, M	LUAD, LUSC	Train
Zilionis et al., 2019	18	37,181	I-IV	F, M	LUAD, LUSC	Test

b

c

Case	Train		Test		Insight
	Class 1	Class 2	Class 1	Class 2	
1	n = 4,852	n = 4,852	n = 3,320	n = 33,861	NSCLC signature
2	n = 3,700	n = 3,700	n = 2,155	n = 27,158	LUAD signature
3	n = 1,152	n = 1,152	n = 1,165	n = 6,703	LUSC signature
4	n = 1,152	n = 1,152	n = 1,165	n = 2,155	Separation between NSCLC subtypes
5	n = 1,152	n = 1,152	n = 2,155	n = 27,158	Across tumor performance
6	n = 3,700	n = 3,700	n = 1,165	n = 6,703	Across tumor performance

LUSC Cancer Cells
 LUAD Cancer Cells
 Healthy Cells

d

e

Figure 6. scMKL demonstrates robust generalization across transfer learning and class imbalance in NSCLC. **a.** Overview of the NSCLC cohorts used for training and testing **b** Impact of class imbalance on classification performance across metrics (AUROC, G-mean, F1-score, recall, and precision) under varying healthy-to-cancer cell ratios and regularization strengths (λ). **c.** Experimental setup for six classification tasks spanning subtype-specific and cross-cohort transfer learning scenarios. **d.** Classification performance of scMKL across all tasks, showing strong generalization between LUAD and LUSC subtypes and across cohorts. **e.** Selected features across tasks, revealing subtype-specific signals such as KRAS signaling in cross-subtype settings and estrogen response in LUAD-trained models.

4. While scMKL's interpretability is a central claim, the manuscript could include more detailed examples of actionable insights derived from interpretability. For instance, showing how selected pathways informed experimental validation.

In response to the reviewer's suggestion, we expanded our analysis of selected pathways in transfer learning experiments to demonstrate the actionable insights derived from scMKL's interpretability. To better support this point, we now include more detailed descriptions in the Results and Discussion sections (see added texts below) and highlight selected pathways from each transfer learning case in Figure 6, showing their overlap with curated oncogenic signatures and previous findings. We believe these examples help illustrate how scMKL's built-in interpretability facilitates biologically meaningful, testable insights beyond classification performance alone.

We added the following text in the results section:

*Lines 1086-1125: "We observed that **TNF α signaling via NF- κ B** and the **reactive oxygen species (ROS)** pathway were consistently co-selected across all transfer learning experiments (Fig 6e). These are tightly interconnected pathways; TNF- α is known to induce ROS production in lung cancer cells, promoting oxidative DNA damage and contributing to tumor progression^{2,3}. Their repeated selection underscores a shared inflammatory and stress-response axis across NSCLC subtypes. Importantly, recent reviews underscore the therapeutic potential of modulating ROS levels as a strategy to eliminate cancer cells by exploiting their metabolic vulnerabilities³.*

***Allograft rejection** was identified in every case except the LUAD vs. LUSC comparison, likely because this pathway reflects broad immune activation and is more relevant for distinguishing tumor from non-tumor cells, rather than between tumor subtypes.*

*The **ER Late pathway** was consistently selected in LUAD-related comparisons (Cases 1, 2, 4, 6), but not in LUSC-specific settings (Cases 3, 5), pointing to LUAD subtype specificity. Supporting this, survival analysis revealed a significant association between ER Late pathway activity and patient outcome in LUAD, while no such association was observed in LUSC ($p < 0.01$ in LUAD; non-significant in LUSC, Figure S7b-c). These results are consistent with prior studies implicating estrogen receptor signaling in LUAD biology⁴ and treatment response⁵, and suggest this pathway may have prognostic and therapeutic relevance in LUAD. The **KRAS signaling** pathway followed a similar pattern, further supporting LUAD-specific regulation which was shown that KRAS mutations have a higher prevalence in LUAD^{6,7}.*

MYC targets were consistently selected in LUSC and in LUAD–LUSC subtype comparisons, but notably absent in LUAD-only classification, indicating its stronger association with LUSC biology and potential as a distinguishing feature. MYC protein expression was elevated in T-LUSC relative to T-LUAD, MYC overexpression induced a “squamous-like” phenotype⁸. Finally, when comparing tumor subtypes (Case 4), scMKL identified additional pathways related to **epithelial–mesenchymal transition (EMT), interferon-alpha response, and metabolism**, suggesting that inter-subtype classification tasks capture broader transcriptional and regulatory variation⁹. These results illustrate scMKL’s ability to mine biologically meaningful and context-specific signals, supporting its utility for mechanistic hypothesis generation and experimental prioritization.”

We added the following text to the Discussion section:

Lines 1096-1211: “To further validate **generalizability**, we applied scMKL to challenging classification tasks involving class imbalance and **cross-cohort prediction** in non-small cell lung cancer (NSCLC). Across six transfer learning scenarios—including cross-subtype classification and cohort transfer—scMKL consistently achieved high performance (Fig. 6). scMKL identified subtype-specific pathways that align with well-characterized mechanisms in non-small cell lung cancer (NSCLC). These findings are not only consistent across train-test studies (**Maroni et al. 2021 and Zilionis et al. 2019, which were collected at different times and processed with different computational pipelines**), but they also illustrate how scMKL highlights pathways with potential therapeutic relevance — such as **Estrogen Response in LUAD and MYC signalling in LUSC** which could **inform drug sensitivity profiling or patient stratification**. These findings highlight the robustness of scMKL across biological and technical variation, including dataset shifts, tumor heterogeneity, and sample imbalance.

By leveraging pathway- and TF-informed kernels, scMKL uncovered **biologically meaningful features associated with patient outcomes**. These interpretable results reinforce the model’s value for **hypothesis generation, therapeutic insight, and prioritization of experimental follow-up or biomarker discovery**.”

Figure S7. Related to main Figure 6 in the main text.

a. Overall pathway selection with varying class imbalance **b-c.** Survival analysis validating that scMKL finding, Hallmark ER Late pathway is LUAD-specific. Differential progression-free interval (PFI) analysis using a Cox proportional-hazards (PH) model for the Hallmark Estrogen Response Late pathway. **Left:** LUAD cohort (Hazard-ratio: 1.95, p-value <0.01). **Right:** LUSC cohort (Hazard ratio: 1.28, p-value 0.29). Hazard ratios and p-values reflect the association between pathway activity and patient outcome.

5. Although scalability metrics are included, discussing computational requirements and reproducibility tools (e.g., code repository availability) would add value to the reader.

We now provide additional metrics and details on computational requirements, including runtime and memory usage across experiments (see Figure 2c-d, Supplementary Figure 2a).

Additionally, to ensure reproducibility, we have made our scalability comparison implementation publicly available, along with scripts to reproduce key analyses and benchmarks.

Lines 1565-1573: Code Availability

An open-source Python implementation of the scMKL is accessible at <https://github.com/ohsu-cedar-comp-hub/scMKL>. The scMKL distribution is also available on the Python Package Index (<https://pypi.org/project/scmkl/>). Results were generated using version 0.1.6 of scMKL. The shiny for python app for visualizing all the results in more detail in this paper are available at https://huggingface.co/spaces/scMKL-team/scMKL_analysis.

Code for comparison algorithms and scalability tests are accessible at https://github.com/ohsu-cedar-comp-hub/scMKL_paper. Results generated by scMKL experiments, with the scripts used to generate them are accessible at [10.5281/zenodo.15397924](https://zenodo.org/record/15397924).

We have added the following text discussing computational requirement comparison between MKL methods in results section :

Lines 306-327: “scMKL addresses these gaps through innovations in both computational efficiency and biological interpretability. It employs Random Fourier Features (RFF) to efficiently approximate nonlinear kernels, reducing computational complexity from $O(N^2)$ to $O(N)$, and uses Group Lasso regularization for sparse, modality-aware feature selection. This design enables scMKL to scale to large, high-dimensional single-cell datasets while maintaining interpretability by identifying the contributions of specific features and modalities.

To further demonstrate scMKL’s advantages, we first benchmarked it against standard kernel-based classifiers, including SVM and EasyMKL. SVM is restricted to a single kernel, while EasyMKL is limited in scalability. In comparisons on our smallest dataset, scMKL achieved comparable classification accuracy while training $7\times$ faster and using $12\times$ less memory than EasyMKL (Fig. 2d). Although we matched the number of pathway-informed kernels for fairness, existing methods became computationally impractical at larger scales. These results, detailed in Figures 2c–d and Supplementary Figure 1a, highlight scMKL’s ability to scale multimodal integration without compromising performance or interpretability.”

Lines 359-362: “Due to the large sample sizes of LUAD and LUSC datasets, it was not feasible to run all state-of-the-art methods without exceeding memory limits. To evaluate computational efficiency, we benchmarked runtime and memory usage with upper bounds of 12h and 50 GB (Fig 2c-d and S2a-b).”

Lines 425-431 and 462-467: “In addition to prediction scores, we compared the scalability of each algorithm by testing them on randomly selected cell samples of different sizes. We showed that for sample sizes approaching thousands of cells, scMKL provided the best AUROC (Fig 2c-d) while scaling better or comparably to the MLP, XGBoost, and SVM algorithms with respect to time and memory (Fig S2a). Furthermore, we compared scMKL with EasyMKL, showing that existing MKL methods do not scale to the size of smaller single-cell datasets. Due to time and memory constraints, this comparison was limited to Hallmark genes (Fig. 2d). We also assessed the effect of dimensionality reduction on performance: We evaluated baseline classifiers (MLP, XGBoost, and SVM) under multiple ATAC input configurations. SVM performance improved with LSI but required substantially more memory and compute time, while MLP and XGBoost were less sensitive to preprocessing choice (Figure S2b). PCA on RNA data did not improve predictive accuracy, while LSI on ATAC data yielded marginal gains at a substantial computational cost [$\sim 2\times$ more time and $\sim 18\times$ more memory] (Fig S2b).”

Lines 1170-1173: “scMKL demonstrates **better or comparable computational efficiency** (Fig. 2c-d, S2a-b), making it both effective and practical for large-scale single-cell studies. Notably, scMKL achieves this with fewer computational resources, due in part to its efficient kernel design and group-aware sparsity constraints.”

Minor Comments:

1. Ensure consistency in terminology, e.g., "multiomics" vs. "multi-omics."

We have standardized terminology throughout the manuscript, including changing all instances of “multi-omics” to “multiomics,” and carefully reviewed the text to ensure consistency in all technical terms.

2. Avoid redundancy, such as in repeated explanations of kernel functions.

We thank the reviewer for pointing this out. While we did not find major redundancy, we made minor edits to improve clarity and avoid potential repetition.

3. Enhance figure captions for clarity; provide brief methods or results within captions (e.g., for Figure 2 and 5).

We now updated figure captions with brief methods and results.

Figure 2. scMKL provides scalable, flexible, and accurate predictions across diverse datasets a-b. Classification performance comparison: MLP, XGBoost, SVM, and scMKL were evaluated across seven datasets, including three multimodal datasets—MCF-7, T-47D, and SLL; and four unimodal datasets—PCa, LUAD, and LUSC scRNA-seq and PCa scATAC-seq. Darker color shades represent models trained on all features, while lighter shades denote models using Hallmark or MVF features for RNA and ATAC. *, **, *** are used to denote statistical significance from Wilcoxon test p values < 0.05 , 0.01 , and 0.001 respectively. Gold indicates

*scMKL had significantly better performance; black indicates the benchmark algorithm had significantly higher performance. c. Comparison of scalability cost: Analysis of training time in seconds and memory in GB across different sample sizes (1k-6k cells) for the four machine learning models (MLP, XGBoost, SVM and scMKL) using all genes, Hallmark genes, Hallmark peaks, and most variable peaks. d. Comparison of MKL methods (EasyMKL vs. scMKL), showing AUROC, memory usage, and runtime as a function of sample size.. e. Comparison of ATAC data groupings and transformations: Predictive performance for binary and TF-IDF normalized ATAC peaks using Hallmark gene sets, Cistrome, and JASPAR TFBS as prior information. *, **, and *** denote significance determined by Wilcoxon tests for p-values < 0.05, 0.01, and 0.001 indication which ATAC data transformation resulted in significantly better AUROC. f–g. Systematic perturbation analysis across RNA and ATAC Hallmark and JASPAR models, measuring performance decline as feature groups were progressively substituted or noise was added to assess biological informativeness.*

Figure 5. scMKL stratifies cells from low- and high-grade prostate cancer tumors identifying factors associated with disease progression and subtypes.

a-b. Classification performance (AUROC, F1-score, precision, recall) between GAS (red), ATAC (blue), and combined GAS+ATAC (purple) models in PCa sc-ATAC-seq data. along with Hallmark pathway selection frequency and kernel importance scores, highlighting consistently selected biological pathways. *, **, and *** denote significance determined by Wilcoxon tests for p-values < 0.05, 0.01, and 0.001 respectively. The color indicates which experiment had significantly higher performance. **c.** AUROC as a function of the sparsity parameter λ (line plot) for scATAC PCa. Also shown are the top 20 selected pathways for each given λ . Shown for GAS Hallmark (red), ATAC Hallmark (blue), and GAS + ATAC Hallmark (purple). **d-e.** Classification performance (AUROC, F1-score, precision, recall) between GAS (red), ATAC (blue), and combined GAS+ATAC (purple) models in PCa sc-ATAC-seq data, along with TF selection frequency and kernel importance scores, highlighting consistently selected biological pathways and TFs. **f.** Hallmark pathway selection frequency and kernel importance scores using an independent scRNA-seq dataset classifying malignant vs non-malignant cells **g.** Selection frequency and kernel importance scores of multimodal NE-differentiation gene sets and JASPAR TFs in classification of G3 vs G4 cells. **h.** The Cistopic UMAP with clusters colored by grade, orange = G3, three shades of pink = G4-1, -2, and -3 clusters. **i.** NE differentiation pathway selection frequency in classification of G3 cells vs individual G4 clusters.

4. Include a detailed legend for Table 1 (dataset information).

We updated the Table 1 and 2 legends as suggested:

Table 1. Summary of Datasets. This table provides an overview of the datasets utilized in scMKL analysis, including sample sizes, feature counts for RNA, ATAC, and GAS data, and the corresponding classification tasks. For each dataset, the number of samples per class is also shown, with class labels indicating the biological comparisons (e.g., control vs. estrogen-treated, healthy vs. tumor, non-malignant vs. malignant). The datasets cover various conditions, including multiomics datasets (RNA+ATAC), single-omics datasets scRNA, and scATAC data, with diverse cancer types, facilitating a comprehensive evaluation of scMKL's performance across different omic layers and biological contexts.

Table 2. Overview of RNA and ATAC Feature Groups. This table provides a summary of the RNA and ATAC feature databases used in scMKL analysis, including Hallmark gene sets and TFBS from Cistrome and JASPAR. For each dataset, the table lists the number of biological groups, total number of features that are in the groupings, number of features per group, number of features exclusive to one group, and number of shared features across groups. Exclusive features refer to features found in only one grouping while shared features refers to features found in multiple groupings. The datasets include multiomics and single-omic single-cell data from various cancer types, allowing for comprehensive analysis of molecular pathways, gene expression, and regulatory elements. The feature group databases support the interpretation of scMKL’s performance across different biological contexts and omic layers.

5. Expand citations for comparison methods and prior information databases to ensure comprehensive coverage.

Thank you for the suggestion. We have expanded the references that highlight the existing approaches and resources commonly used.

6. Specify which pathway enrichment tests were used in benchmarking scMKL against standard workflows (e.g., GSEA).

We thank the reviewer for this suggestion. We have now clarified the enrichment methods used in the benchmarking. Specifically, Enrichr applies Fisher’s exact test with a combined score for ranking, while GSEA Preranked uses a Kolmogorov–Smirnov-like statistic with permutation testing of gene sets. This information has been added to the Methods section. Lines 1519-1521.

7. Define all acronyms at first use and include a table for quick reference (e.g., AUROC, TF-IDF).

We now have an abbreviations table.

Table S1: Abbreviations and Definitions

Abbreviation	Definition
AR	Androgen Response
ATAC	Assay for Transposase-Accessible Chromatin
AUROC	Area Under the Receiver Operating Characteristic curve
DEG	Differentially Expressed Gene
ER	Estrogen Response
GAS	Gene Accessibility Score
GL	Group Lasso
GREAT	Genomic Regions Enrichment of Annotations Tool
GSEA	Gene Set Enrichment Analysis
LUAD	Lung Adenocarcinoma
LUSC	Lung Squamous Cell Carcinoma
LSI	Latent Semantic Indexing
MKL	Multiple Kernel Learning
MLP	Multi-layer Perceptron
NE	Neuroendocrine

NSCLC	Non-Small Cell Lung Cancer
PCA	Principle Component Analysis
PCa	Prostate Cancer
RFF	Random Fourier Features
RNA-seq	Sequencing of Messenger Ribonucleic Acid
sc	Single-Cell
scMKL	Single-Cell Multiple Kernel Learning
SLL	Small Lymphocytic Lymphoma
TCGA	Tumor Cancer Genome Atlas
TF	Transcription Factor
TFBS	Transcription Factor Binding Sites
TF-IDF	Term Frequency-Inverse Document Frequency
UMAP	Uniform Manifold Approximation and Projection

The manuscript is an impressive contribution to the field, offering a well-rounded, interpretable solution for analyzing complex single-cell data. With enhancements to clarity, expanded reproducibility discussions, and deeper examples of interpretability, this work will be a strong candidate for publication in computational biology or systems biology journals.

Thank you for your kind and constructive feedback. We are grateful for your recognition of our work as an important contribution to the field. We appreciate your suggestions for further enhancing the clarity, reproducibility discussions, and interpretability examples in the manuscript. In response to your comments, we have:

1. Detailed comparison with existing kernel-based methods. We expanded our discussion of scMKL's advantages over existing kernel-based methods, including empirical comparisons with both SVM (with RBF kernel) and EasyMKL.

2. Interpretability . To provide a deeper example of interpretability, we benchmarked scMKL across transfer learning scenarios to test its generalization within and across tumor subtypes. These results demonstrate scMKL's robustness and highlight biologically meaningful features linked to patient outcomes, reinforcing its value for hypothesis generation therapeutic insight, and prioritization of experimental follow-up or biomarker discovery.

3. Computational requirements; We included new figures and expanded sections describing the computational requirements of scMKL, including memory usage and training time. Finally, we provide the code for the scalability comparisons at https://github.com/ohsu-cedar-comp-hub/scMKL_paper in addition to scMKL's github at <https://github.com/ohsu-cedar-comp-hub/scMKL>.

We believe these improvements strengthen the manuscript. We thank the reviewer again for the valuable feedback, which has helped refine the presentation of our work.

- 1 Bredikhin, D., Kats, I. & Stegle, O. MUON: multimodal omics analysis framework. *Genome Biol* **23**, 42, doi:10.1186/s13059-021-02577-8 (2022).
- 2 ArulJothi, K. N. *et al.* Implications of reactive oxygen species in lung cancer and exploiting it for therapeutic interventions. *Med Oncol* **40**, 43, doi:10.1007/s12032-022-01900-y (2022).
- 3 Perillo, B. *et al.* ROS in cancer therapy: the bright side of the moon. *Exp Mol Med* **52**, 192-203, doi:10.1038/s12276-020-0384-2 (2020).
- 4 Jia, S. *et al.* Transcriptome Based Estrogen Related Genes Biomarkers for Diagnosis and Prognosis in Non-small Cell Lung Cancer. *Front Genet* **12**, 666396, doi:10.3389/fgene.2021.666396 (2021).
- 5 Wang, Y. *et al.* Immune characteristics analysis and construction of a four-gene prognostic signature for lung adenocarcinoma based on estrogen reactivity. *Bmc Cancer* **23**, 1047, doi:10.1186/s12885-023-11415-y (2023).
- 6 Shen, Y., Chen, J. Q. & Li, X. P. Differences between lung adenocarcinoma and lung squamous cell carcinoma: Driver genes, therapeutic targets, and clinical efficacy. *Genes Dis* **12**, 101374, doi:10.1016/j.gendis.2024.101374 (2025).
- 7 Acker, F. *et al.* KRAS Mutations in Squamous Cell Carcinomas of the Lung. *Front Oncol* **11**, 788084, doi:10.3389/fonc.2021.788084 (2021).
- 8 Quintanal-Villalonga, A. *et al.* Comprehensive molecular characterization of lung tumors implicates AKT and MYC signaling in adenocarcinoma to squamous cell transdifferentiation. *J Hematol Oncol* **14**, 170, doi:10.1186/s13045-021-01186-z (2021).
- 9 Li, F. *et al.* Comprehensive analysis of the role of a four-gene signature based on EMT in the prognosis, immunity, and treatment of lung squamous cell carcinoma. *Aging (Albany NY)* **15**, 6865-6893, doi:10.18632/aging.204878 (2023).
- 10 Garcia-Martinez, L., Zhang, Y., Nakata, Y., Chan, H. L. & Morey, L. Epigenetic mechanisms in breast cancer therapy and resistance. *Nat Commun* **12**, 1786, doi:10.1038/s41467-021-22024-3 (2021).

Note: Reviewer comments are in **blue**, our responses are in **black**, and text added to the revised manuscript is *italicized*.

Reviewer #2 (Remarks to the Author):

This manuscript presents a method based on scMKL, designed for integrative single-cell multiomics analysis using RNA and ATAC data. The authors demonstrate the potential of this method in improving classification accuracy and interpreting biological phenomena. The study offers significant innovation and methodological advancement. However, while the paper achieves notable results in multi-modal data integration and model application, there remain several issues that need addressing, particularly regarding interpretability, computational efficiency, generalization ability, and comparisons of preprocessing strategies. Below are specific reviewer concerns.

interpretability → created new figures, updated text

computational efficiency → we discuss efficiency in more detail with added comparisons

generalization ability → we added a new dataset where we use transfer learning in different settings

comparisons of preprocessing → we added a new figure comparing all preprocessing algorithms and their effects to prediction performance

1. Interpretability of Multi-Modal Data Integration and Model Transparency

Concern: The paper emphasizes the advantages of scMKL in integrating multi-modal data (RNA + ATAC), but the interpretability of the integration process remains somewhat unclear. While the authors demonstrate improved classification accuracy, they provide limited explanation regarding how the model handles the interaction between different modalities. Specifically, how the model avoids information loss or redundancy between modalities and ensures integration does not lead to overfitting is not sufficiently discussed.

Recommendation: The authors should elaborate on the specific strategies used in the scMKL model for handling multi-modal data, especially on how biological information is effectively captured during integration. Additionally, a more detailed discussion on model transparency and interpretability is needed, such as explaining the decision-making process, feature importance, etc. Further visualization techniques (e.g., heatmaps, interpretable model diagrams) would help readers better understand how different data modalities influence the final decision of the model.

We appreciate the reviewer's insightful comments regarding the interpretability of multi-modal integration in scMKL. To clarify, scMKL avoids information loss and redundancy by constructing modality-specific kernels informed by biological priors—specifically, pathway-defined gene sets for RNA and transcription factor (TF)-associated regions for ATAC. Rather than concatenating features early, scMKL integrates modalities at the decision level, allowing each to contribute independently and preserving modality-specific biological signals.

To enhance transparency, we systematically evaluate feature importance via kernel weights and selection frequency across 100 independent train-test splits. This analysis reveals which

pathways and TFs are consistently informative, supporting model reproducibility and interpretability. We now include a dot plot summarizing this analysis, where dot size indicates selection frequency and color intensity reflects kernel weight magnitude.

We have revised Figure 1 (see below) to include a schematic summarizing scMKL's interpretable integration process and model selection. We also expanded the visualization of model outputs, showing selection stability and weighted contributions of biological feature groups.

Finally, to further demonstrate the model's reliance on meaningful biological structure, we added ablation and perturbation studies. These analyses show that scMKL's performance and feature selection degrade when domain-informed kernels are randomized or removed—highlighting the importance of biologically guided integration.

These revisions aim to clarify how scMKL integrates multi-modal data in a biologically interpretable and robust manner, as reflected in the updated figures (see figure 1 below and other figures are updated with manuscript text and figures.)

Lines 175-193: “To provide a transparent and inherently interpretable framework for analyzing single-cell multiomics data, we developed scMKL, which integrates both expert knowledge and RNA and ATAC modalities using biologically informed kernel functions (Fig. 1a). In the multi-view setting, scMKL incorporates pathway-based gene groups for RNA-seq and TFBS-associated regions for ATAC-seq to construct modality-specific kernels. Each kernel is approximated using random Fourier features (Z_i) and assigned an importance weight (η_i) during model training. This framework preserves modality-specific information while allowing their integration at the decision level. Interpretability is achieved by visualizing feature group selection frequency and kernel weight contributions using dot plots, enabling biological insight into which pathways or TFs drive classification.

To ensure robustness and avoid overfitting, scMKL employs an 80/20 train-test split repeated 100 times with varying random seeds, combined with 4-fold cross-validation within each training set to optimize the regularization parameter λ (Fig. 1b). This nested evaluation framework supports consistent model performance across different data partitions and enhances reproducibility.

We further examined how λ influences the sparsity of feature selection (Fig. 1c). As expected, stronger regularization (higher λ) results in fewer selected pathways ($\eta_i \neq 0$), increasing model sparsity and interpretability while reducing potential overfitting. Conversely, lower λ values allow the model to capture more biological variation but may compromise generalizability. This analysis highlights the tunable balance in scMKL between interpretability and biological breadth.”

a

scMKL

Figure 1. Interpretable machine learning framework for single-cell multiomics data analysis using scMKL.

a. Multi-view and multi-modal integration in scMKL. scMKL integrates single-cell RNA-seq and ATAC-seq data with expert-curated biological knowledge to enhance interpretability and performance. In the multi-view setting, modality-specific kernels are constructed from gene sets (RNA) and TF-binding regions (ATAC), allowing distinct biological signals to be captured independently. The kernel approximation Z and learned weight η reflect the importance of each feature group in classification. Interpretability is achieved through dot plots showing selection frequency and weight of each pathway or TF group.

b. Cross-validation and robust evaluation. An outer 80/20 train-test split is repeated 100 times with different random seeds. Within each training set, 4-fold cross-validation is used to tune the regularization parameter λ . This nested setup ensures robust generalization and guards against overfitting.

c. Effect of regularization on pathway sparsity. Increasing λ leads to more sparsity in feature selection, reducing the number of pathways or TFs retained. This panel illustrates the trade-off

between interpretability (fewer, more informative features) and potential biological coverage, underscoring the importance of tuning λ appropriately.

Line 534-602: “To examine whether the accuracy stemmed from prior information groupings that effectively capture biologically relevant features, we conducted a pathway perturbation analysis on the MCF-7 dataset where we randomized a portion of pathway features or added noise features instead in a second experiment (Fig 2f-g). We expected that the more biologically informative the prior information groupings were, the higher the AUROC would be. As we progressively increased the randomness or added noise within the feature groups, respectively, AUROC values decreased as expected in each case. This decline was most evident for RNA Hallmark, TF-IDF ATAC Hallmark, binary ATAC Cistrome, and Binary ATAC JASPAR groupings, suggesting these strategies are particularly effective in capturing meaningful biological mechanisms for MCF-7.”

Line 624-664: “We assessed which feature groups scMKL consistently selected across varying solution sparsity levels (i.e., feature groups scMKL used for prediction) (Fig 3c). Figure 3c, shows the Hallmark pathway and JASPAR TF selection in unimodal settings—using only RNA data or ATAC data and in multimodal setting—using the combination of the two data modalities. Combination of RNA and ATAC gives the best prediction scores, especially in terms of F1-score, precision, and recall (Fig 3a). We quantified the informativeness of each Hallmark pathway and JASPAR TF in the classification of MCF-7 cells control versus estrogen treated by the selection frequency and the kernel weight using RNA, ATAC, and RNA+ATAC multiomics data integration (Fig 3b). In the JASPAR-based ATAC data analysis, ESR1 and ESR2 were the most frequently selected TFs (Fig 3b, c). ESR1 (ER α) is the primary mediator of estrogen signaling in breast cancer, regulating genes involved in cell cycle progression, while ESR2 (ER β) can act as a modulator of ER α activity and has context-dependent roles in differentiation and tumor suppression¹. The selection frequency for Hallmark pathways and JASPAR TFs using RNA and ATAC data combined, showed a strong representation of the Estrogen Response (ER) Early RNA pathway, and ESR1 ATAC TF groups. Overall, ER Early and ESR1 TF groups are consistent across all panels, underscoring their importance in the estrogen-treated condition. Combined RNA and ATAC data reinforced the prominence of these features.

To test the functional importance of top selected feature groups, we performed ablation studies by excluding top-ranked unimodal and multimodal feature groups from training; in all cases, model performance decreased, confirming that scMKL not only selects biologically relevant features but also relies on them for accurate classification (Fig 3d). Interestingly, when either ESR1/ESR2 or the ER Early/Late pathways was removed from training, scMKL adaptively up-weighted the remaining estrogen-related features (Figure 3e), indicating redundancy within the estrogen signaling axis and the model’s ability to re-prioritize biologically relevant groups.”

Figure 3. scMKL enables accurate, interpretable multimodal classification and identifies biologically relevant mechanisms through robust feature selection.

a. Comparison of classification performance (AUROC, F1-score, precision, recall) between RNA (red), ATAC (blue), and RNA+ATAC (purple) models across MCF-7, T-47D, and SLL multimodal datasets. *, **, and *** denote significance determined by Wilcoxon tests for p-values < 0.05, 0.01, and 0.001 respectively. The color indicates which experiment had significantly higher performance. **b.** Feature group selection frequency and kernel importance scores across RNA, ATAC, and RNA+ATAC models, highlighting consistently selected biological pathways and TFs. **c.** AUROC and feature group selection frequency across RNA, ATAC, and RNA+ATAC models, identifying critical feature groups consistently across different regularization values (λ). **d-e.** Ablation (leave-one-group-out) analysis showing the impact on AUROC and the importance of the other feature groups after removal of selected feature groups (e.g., ER early/late response, ESR1/ESR2) in the MCF-7 dataset compared to baseline RNA+ATAC model. *, **, and *** denote significance determined by Wilcoxon tests for p-values < 0.05, 0.01, and 0.001 indication which ablation experiments resulted in significantly worse AUROC than the baseline. **f-g.** Systematic perturbation analysis across RNA and ATAC Hallmark and JASPAR models, measuring performance decline as feature groups were progressively substituted or noise was added to assess biological informativeness.

2. Computational Efficiency of ATAC Feature Selection

Concern: The paper mentions that the high-dimensionality of ATAC data limits computational resources, and thus only the most variable ATAC features are used for analysis. However, the process of ATAC feature selection could affect the model's overall performance, particularly under resource constraints. While the authors claim that scMKL outperforms traditional classifiers in these conditions, the balance between computational resource constraints and classification accuracy is not thoroughly explored.

Recommendation: The authors should consider optimizing the ATAC feature selection process, potentially by introducing more efficient feature selection strategies or dimensionality reduction methods. Techniques like PCA, t-SNE, or sparse feature selection methods (e.g., LASSO) could help improve computational efficiency while retaining sufficient biological information. Additionally, comparative experiments assessing the impact of different feature selection strategies under resource-limited conditions would be beneficial.

We appreciate the reviewer's suggestion regarding ATAC feature selection. In our approach, feature selection is guided by biological priors, focusing on constructing feature groups based on peaks overlapping known gene regions, transcription factor binding sites, or curated pathways. This biologically informed grouping ensures interpretability while reducing redundancy and dimensionality in a principled manner.

To further evaluate the balance between computational efficiency and classification performance, we conducted new experiments comparing multiple ATAC preprocessing strategies: (1) binary peak counts, (2) TF-IDF transformation, (3) Latent Semantic Indexing (LSI), and (4) a standard Scanpy workflow with highly variable peak selection.

For scMKL, performance remained robust across all preprocessing strategies, with minor variation in accuracy and no significant increase in runtime, demonstrating the framework's resilience to different ATAC representations. We also tested baseline classifiers (SVM, MLP,

XGBoost) using LSI, Hallmark peak sets, and most variable peaks. MLP and XGBoost showed little sensitivity to preprocessing choice, while SVM performance improved with LSI but at the cost of substantially increased memory and compute time as sample size grew.

These results support that scMKL achieves a favorable trade-off between interpretability, performance, and computational cost. We have added these findings to the manuscript along with a new supplementary figure summarizing the results.

Figure S3. Comparison of preprocessing strategies for RNA and ATAC feature representations across multiple datasets b. ATAC modality: AUROC, F1-score, precision, and recall for classifiers using ATAC peak counts from the same datasets. Comparisons include binary peak presence/absence matrix, TF-IDF transformation, and LSI via TruncatedSVD (50 components, first component removed), as well as the standard workflow using most variable peaks.

Figure S2. scMKL provides scalable, flexible, and accurate predictions across diverse datasets and identifies biologically relevant mechanisms through pathway/TF selection

b. Comparison of scalability cost including dimensionality reduction methods, PCA for RNA and LSI for ATAC: Analysis of training time in seconds and memory in GB across different sample sizes (1k-6k cells) for all the four machine learning models (MLP, XGBoost, SVM, and scMKL) using all genes, Hallmark genes, Hallmark peaks, and most variable peaks.

The following text added to results and discussion section:

Lines462-409: “We assessed the effect of dimensionality reduction on performance: We evaluated baseline classifiers (MLP, XGBoost, and SVM) under multiple ATAC input configurations. SVM performance improved with LSI but required substantially more memory and compute time, while MLP and XGBoost were less sensitive to preprocessing choice (Figure S2b). PCA on RNA data did not improve predictive accuracy, while LSI on ATAC data yielded marginal gains at a substantial computational cost [$\sim 2\times$ more time and $\sim 18\times$ more memory] (Fig S2b).

To evaluate the impact of different preprocessing strategies on scMKL, we conducted a series of experiments focused on gene selection, data normalization, and dimensionality reduction. scMKL does not rely on PCA or LSI prior to modeling. Instead, it leverages biologically informed groupings (e.g., pathways/gene sets or TF-informed peak groups), allowing for structured feature construction and selection. However, we benchmarked several preprocessing alternatives for both RNA and ATAC after construction of each feature group. For RNA, we compared PCA, log-normalization, raw counts, and standard scRNA-seq preprocessing workflow (i.e., using scanpy, including filtering low-quality cells and genes, normalization,

dimensionality reduction). For ATAC, we compared TF-IDF, LSI, binary counts, standard scATAC-seq preprocessing workflow (i.e., using scanpy/muon suite², including filtering low-quality cells and peaks, normalization, dimensionality reduction). scMKL demonstrated robust performance across a range of preprocessing methods, with metrics AUROC, F1-Score, Precision, and Recall consistently achieving high values (e.g., AUROC spanning from 0.9731 to 0.9999). Notably, the optimal preprocessing method varied depending on both the dataset and the specific performance metric, highlighting that while scMKL was consistently effective across different approaches, no single preprocessing technique proved universally superior. This variability underscores the adaptability of scMKL, ensuring reliable outcomes across diverse biological datasets and preprocessing strategies. Furthermore, filtering cells effectively reduces ambiguity, thereby enhancing data clarity between distinct classes. In this context, after standard processing workflows, the Binary method performed best across T47D, Lymphoma, and Prostate, while LSI outperformed the other methods in MCF7 (Fig S3).

Together, these advances establish scMKL as an efficient and biologically grounded interpretation tool without sacrificing performance, offering a practical advantage in resource-limited settings for multimodal single-cell data integration.”

3. Model Generalization and Cross-Dataset Validation

Concern: Although the authors demonstrate scMKL's superior performance across different cell lines and datasets, more validation is needed to assess its generalization ability across datasets. Specifically, it remains unclear whether the model can maintain high performance across different tumor types and cell types, such as comparing data from prostate cancer versus breast cancer cell lines.

Recommendation: The authors should further validate scMKL's generalization capability, particularly on a broader range of cancer datasets. Testing the model on different tumor types and clinical backgrounds would help assess its robustness and performance consistency across datasets. Exploring ways to optimize the model to handle diverse tumor types would further enhance its clinical applicability.

We appreciate the reviewer's thoughtful suggestion regarding cross-dataset generalization. To address this, we evaluated scMKL on a set of **challenging cross-cohort and cross-subtype classification tasks in non-small cell lung cancer (NSCLC)**, leveraging two independent studies (GSE136246 and GSE127465) that differ in collection time and computational processing pipelines, and sample composition (multiple patients with Lung Adenocarcinoma (LUAD) and Lung Squamous Cell Carcinoma (LUSC)).

We designed six transfer learning scenarios, including **cross-subtype (LUAD vs. LUSC) and cross-cohort prediction**, and found that scMKL maintained high classification accuracy in all settings (Fig. 6). Importantly, **scMKL identified subtype-specific pathways**, such as *Estrogen Response* in LUAD and *Myc Signalling* in LUSC, demonstrating that the model captures biologically relevant signals beyond dataset-specific artifacts.

These findings highlight scMKL's ability to generalize across datasets with **biological and technical variation**, reinforcing its applicability to diverse tumor types. We have updated the Results and Discussion sections accordingly to clarify this point and emphasize scMKL's robustness in cross-dataset settings.

We added a new section "scMKL generalizes across tumor subtypes and cohorts while handling class imbalance"

Lines 1065-1144: To evaluate scMKL's generalization across tumor subtypes, cohorts, and technical variation, we benchmarked its performance on six classification tasks using two independent NSCLC scRNA-seq datasets (GSE136346 and GSE127465), which include samples from both lung adenocarcinoma (LUAD) and lung squamous cell carcinoma (LUSC) (Fig. 6a, c).

Dataset construction and evaluation strategy: *We organized datasets by cancer subtype to generate separate LUAD and LUSC cohorts. scMKL was trained on Maroni et al. (2021) and tested on Zilionis et al. (2019) across all tasks. To control for class imbalance, healthy cells were subsampled in the training set.*

Classification tasks:

1. *Distinguishing healthy cells from cancer cells using pooled LUAD and LUSC data.*
2. *Classifying healthy vs. cancer cells within LUAD only.*
3. *Classifying healthy vs. cancer cells within LUSC only.*
4. *Differentiating LUAD from LUSC tumor cells.*
5. *Training on LUSC and testing on LUAD (healthy vs. cancer).*
6. *Training on LUAD and testing on LUSC (healthy vs. cancer).*

Robustness to class imbalance: *To assess the effect of class imbalance, we varied the healthy-to-cancer ratio in training and evaluated classification performance across ten replicates per condition (Fig 6b). scMKL remained stable across a wide imbalance range (Fig 6b, S7a). Performance improved with greater feature group inclusion (lower regularization) under skewed ratios, while moderate sparsity (higher λ) yielded optimal results near balance (0.6-1.6).*

We observed that TNF α signaling via NF- κ B and the reactive oxygen species (ROS) pathway were consistently co-selected across all transfer learning experiments (Fig 6e). These are tightly interconnected pathways; TNF- α is known to induce ROS production in lung cancer cells, promoting oxidative DNA damage and contributing to tumor progression^{3,4}. Their repeated selection underscores a shared inflammatory and stress-response axis across NSCLC subtypes. Importantly, recent reviews underscore the therapeutic potential of modulating ROS levels as a strategy to eliminate cancer cells by exploiting their metabolic vulnerabilities⁴.

Allograft rejection was identified in every case except the LUAD vs. LUSC comparison, likely because this pathway reflects broad immune activation and is more relevant for distinguishing tumor from non-tumor cells, rather than between tumor subtypes.

The ER Late pathway was consistently selected in LUAD-related comparisons (Cases 1, 2, 4, 6), but not in LUSC-specific settings (Cases 3, 5), pointing to LUAD subtype specificity. Supporting this, survival analysis in TCGA revealed a significant association between ER Late pathway activity and patient outcome in LUAD, while no such association was observed in LUSC

(p < 0.01 in LUAD; non-significant in LUSC, Figure S7b-c). These results are consistent with prior studies implicating estrogen receptor signaling in LUAD biology⁵ and treatment response⁶, and suggest this pathway may have prognostic and therapeutic relevance in LUAD. The KRAS signaling pathway followed a similar pattern, further supporting LUAD-specific regulation which was shown that KRAS mutations have a higher prevalence in LUAD^{7,8}.

MYC targets were consistently selected in LUSC and in LUAD–LUSC subtype comparisons, but notably absent in LUAD-only classification, indicating its stronger association with LUSC biology and potential as a distinguishing feature. MYC protein expression was elevated in T-LUSC relative to T-LUAD, MYC overexpression induced a “squamous-like” phenotype⁹.

Finally, when comparing tumor subtypes (Case 4), scMKL identified additional pathways related to epithelial–mesenchymal transition (EMT), interferon-alpha response, and metabolism, suggesting that inter-subtype classification tasks capture broader transcriptional and regulatory variation¹⁰. These results illustrate scMKL’s ability to mine biologically meaningful and context-specific signals, supporting its utility for mechanistic hypothesis generation and experimental prioritization.”

a

Dataset	Sample #	QC-passed cell #	Stage	Gender	NSCLC subtype	Use of dataset
Maroni et al., 2021	24	53,190	I-IV	F, M	LUAD, LUSC	Train
Zilionis et al., 2019	18	37,181	I-IV	F, M	LUAD, LUSC	Test

b

c

Case	Train		Test		Insight
	Maroni et al., 2021	Maroni et al., 2021	Zilionis et al., 2019	Zilionis et al., 2019	
	Class 1	Class 2	Class 1	Class 2	
1	n = 4,852	n = 4,852	n = 3,320	n = 33,861	NSCLC signature
2	n = 3,700	n = 3,700	n = 2,155	n = 27,158	LUAD signature
3	n = 1,152	n = 1,152	n = 1,165	n = 6,703	LUSC signature
4	n = 1,152	n = 1,152	n = 1,165	n = 2,155	Separation between NSCLC subtypes
5	n = 1,152	n = 1,152	n = 2,155	n = 27,158	Across tumor performance
6	n = 3,700	n = 3,700	n = 1,165	n = 6,703	Across tumor performance

LUSC Cancer Cells
 LUAD Cancer Cells
 Healthy Cells

d

e

Figure 6. scMKL demonstrates robust generalization across transfer learning and class imbalance in NSCLC.

a. Overview of the NSCLC cohorts used for training and testing **b.** Impact of class imbalance on classification performance across metrics (AUROC, G-mean, F1-score, recall, and precision) under varying healthy-to-cancer cell ratios and regularization strengths (λ). **c.** Experimental setup for six classification tasks spanning subtype-specific and cross-cohort transfer learning scenarios. **d.** Classification performance of scMKL across all tasks, showing strong generalization between LUAD and LUSC subtypes and across cohorts. **e.** Selected features across tasks, revealing subtype-specific signals such as MYC signaling in cross-subtype settings and ER Late in LUAD-trained models.

Figure S7. Related to main Figure 6 in the main text.

b-c. Survival analysis validating that scMKL finding in TCGA, Hallmark ER Late pathway is LUAD-specific. Differential progression-free interval (PFI) analysis using a Cox proportional-hazards (PH) model for the Hallmark Estrogen Response Late pathway. **Left:** LUAD cohort (Hazard-ratio: 1.95, p-value <0.01). **Right:** LUSC cohort (Hazard ratio: 1.28, p-value 0.29). Hazard ratios and p-values reflect the association between pathway activity and patient outcome.

4. Comparison and Impact of Different Preprocessing Strategies

Concern: The manuscript compares scMKL with standard single-cell analysis workflows (such as Seurat and scanpy) and highlights the impact of different preprocessing strategies on model performance. However, preprocessing steps can significantly influence the final results, and there is insufficient discussion regarding how the choice of genes, dimensionality reduction, and differential analysis steps affects the outcome.

Recommendation: A more in-depth discussion of how different preprocessing steps (e.g., gene selection, dimensionality reduction, data normalization) influence the performance of the scMKL model is needed. Additionally, providing more experimental data comparing different preprocessing strategies and their effects on model performance would help readers better understand the importance of preprocessing choices in this context.

We appreciate the reviewer’s insightful comment regarding the role of preprocessing in shaping model performance. We addressed preprocessing for ATAC in the previous comment, we also performed similar experiments with RNA as well detailed as follows: *“To evaluate the impact of different preprocessing strategies on scMKL, we conducted a series of experiments focused on gene selection, data normalization, and dimensionality reduction. scMKL does not rely on PCA or LSI prior to modeling. Instead, it leverages biologically informed groupings (e.g., pathways/gene sets or TF-informed peak groups), allowing for structured feature construction and selection. However, we benchmarked several preprocessing alternatives for both RNA and ATAC after construction of each feature group. For RNA, we compared PCA, log-normalization, raw counts, and standard scRNA-seq preprocessing workflow (i.e., using scanpy, including filtering low-quality cells and genes, normalization, dimensionality reduction). For ATAC, we compared TF-IDF, LSI, binary counts, standard scATAC-seq preprocessing workflow (i.e., using scanpy/muon suite², including filtering low-quality cells and peaks, normalization, dimensionality reduction). scMKL demonstrated robust performance across a range of preprocessing methods, with metrics AUROC, F1-Score, Precision, and Recall consistently achieving high values (e.g., AUROC spanning from 0.9731 to 0.9999). Notably, the optimal preprocessing method varied depending on both the dataset and the specific performance metric, highlighting that while scMKL was consistently effective across different approaches, no single preprocessing technique proved universally superior. This variability underscores the adaptability of scMKL, ensuring reliable outcomes across diverse biological datasets and preprocessing strategies. Furthermore, filtering cells effectively reduces ambiguity, thereby enhancing data clarity between distinct classes. In this context, after standard processing workflows, the Binary method performed best across T47D, Lymphoma, and Prostate, while LSI outperformed the other methods in MCF7 (Fig S3).*

Together, these advances establish scMKL as an efficient and biologically grounded interpretation tool without sacrificing performance, offering a practical advantage in resource-limited settings for multimodal single-cell data integration.” Lines 469-509.

Lines 462-467: “We assessed the effect of dimensionality reduction on performance: We evaluated baseline classifiers (MLP, XGBoost, and SVM) under multiple ATAC input configurations. SVM performance improved with LSI but required substantially more memory and compute time, while MLP and XGBoost were less sensitive to preprocessing choice (Figure S2b). PCA on RNA data did not improve predictive accuracy, while LSI on ATAC data yielded marginal gains at a substantial computational cost [$\sim 2\times$ more time and $\sim 18\times$ more memory] (Fig S2b).”

These results reinforce scMKL’s flexibility and its robustness to variations in preprocessing steps, which we now discuss more clearly in the revised Methods and Results sections.

In parallel, we improved the quality of TFBS groupings by updating the reference database (JASPAR 2024, replacing 2022) and constructing groupings based on unique TF IDs rather than names, which reduced redundancy and improved interpretability. Using MCF-7 ATAC data, we benchmarked groupings based on the top 1,000, 2,000, and 5,000 peaks by motif match score, as well as groupings including all peaks matched to TF motifs. Across all comparisons, the updated high-confidence groupings improved both model accuracy and biological interpretability (Figure

S8). These improvements collectively underscore our effort to ensure that scMKL remains both computationally tractable and biologically informative, even under diverse preprocessing configurations.

We added the following text:

Lines 1170-1176: “We also showed scMKL’s flexibility and its robustness to variations in preprocessing steps (Figure S3). In addition, scMKL demonstrates better or comparable computational efficiency (Fig. 2c-d, S2a-b), making it both effective and practical for large-scale single-cell studies. Notably, scMKL achieves this with fewer computational resources, due in part to its efficient kernel design and group-aware sparsity constraints. We benchmarked runtime and memory usage under varying feature-group sizes, revealing that scMKL remains scalable and efficient even when integrating large prior knowledge sets, such as JASPAR TFs (Fig S8).”

Figure S3. Comparison of preprocessing strategies for RNA and ATAC feature representations across multiple datasets **a.** *RNA modality:* AUROC, F1-score, precision, and recall for scMKL trained on RNA data from MCF7, T47D, SLL, and prostate cancer datasets. Comparisons include: PCA on log-normalized and z-scored RNA counts (50 components), standard workflow using all genes, and log(z)-transformed counts without dimensionality reduction. **b.** *ATAC modality:* AUROC, F1-score, precision, and recall for classifiers using ATAC peak counts from the same datasets. Comparisons include binary peak presence/absence matrix, TF-IDF transformation, and LSI via TruncatedSVD (50 components, first component removed), as well as the standard workflow using most variable peaks.

Figure S8. TFBS-informed grouping strategy comparisons.

a. Classification accuracy comparison of scMKL using MCF-7 ATAC data across four versions of TFBS-informed peak sets: top 1K, 2K, 5K most significant peaks, and all. Matched peaks from the JASPAR 2024 database across different regularization levels λ . **b.** Memory and time usage for each peak grouping strategy, showing that limiting peak sets (1k and 2k especially) substantially reduces computational cost while maintaining or improving performance. **c.** Heatmaps showing selected TF groups across different regularization (λ). Overlap in selected TFs across versions, emphasizing that high-confidence peak selection preserves core regulatory signals.

5. Minor Formatting Issues

Concern: There are some formatting inconsistencies in the paper, such as differences in font between Table 1 and Table 2. Additionally, in Figure 2 and other figures, boxplots lack statistical significance annotations, which are essential for guiding the reader on whether the observed differences between methods are statistically significant.

Recommendation: The authors should standardize the font across tables and figures for consistency. Furthermore, statistical significance annotations should be included in the boxplots in Figure 2 and other relevant figures to improve clarity and help readers assess the statistical validity of the reported differences.

Thank you for pointing this out. We have carefully reviewed and corrected the formatting inconsistencies between Table 1 and Table 2 to ensure consistency in font style and layout throughout the manuscript. Additionally, we have updated Figure 2 and all relevant boxplots to include statistical significance annotations (e.g., $p < 0.05$, $p < 0.01$, $p < 0.001$) using appropriate statistical tests (e.g., Wilcoxon signed-rank test) to highlight meaningful differences between methods. These changes improve clarity and allow readers to better assess the performance comparisons. We have updated the following figure panels:

Figure 2. scMKL provides scalable, flexible, and accurate predictions across diverse datasets a-b. *Classification performance comparison:* MLP, XGBoost, SVM, and scMKL were evaluated across seven datasets, including three multimodal datasets—MCF-7, T-47D, and SLL; and four unimodal datasets—PCa, LUAD, and LUSC scRNA-seq and PCa scATAC-seq. Darker color shades represent models trained on all features, while lighter shades denote models using Hallmark or MVF features for RNA and ATAC. *, **, *** are used to denote statistical significance from Wilcoxon test p values < 0.05, 0.01, and 0.001 respectively. Gold indicates scMKL had significantly better performance; black indicates the benchmark algorithm had significantly higher performance. **e.** *Comparison of ATAC data groupings and transformations:* Predictive performance for binary and TF-IDF normalized ATAC peaks using Hallmark gene sets, Cistrome, and JASPAR TFBS as prior information. *, **, and *** denote significance

determined by Wilcoxon tests for p-values < 0.05, 0.01, and 0.001 indicating which ATAC data transformation resulted in significantly better AUROC.

Figure 3. scMKL enables accurate, interpretable multimodal classification and identifies biologically relevant mechanisms through robust feature selection.

a. Comparison of classification performance (AUROC, F1-score, precision, recall) between RNA (red), ATAC (blue), and RNA+ATAC (purple) models across MCF-7, T-47D, and SLL multimodal datasets. *, **, and *** denote significance determined by Wilcoxon tests for p-values < 0.05, 0.01, and 0.001 respectively. The color indicates which experiment had significantly higher performance. **d-e.** Ablation (leave-one-group-out) analysis showing the impact on AUROC and the importance of the other feature groups after removal of selected feature groups (e.g., ER early/late response, ESR1/ESR2) in the MCF-7 dataset compared to baseline RNA+ATAC model. *, **, and *** denote significance determined by Wilcoxon tests for p-values < 0.05, 0.01, and 0.001 indicating which ablation experiments resulted in significantly worse AUROC than the baseline.

Figure 5. scMKL stratifies cells from low- and high-grade prostate cancer tumors identifying factors associated with disease progression and subtypes.

a-b. Classification performance (AUROC, F1-score, precision, recall) between GAS (red), ATAC (blue), and combined GAS+ATAC (purple) models in PCa sc-ATAC-seq data. along with Hallmark pathway selection frequency and kernel importance scores, highlighting consistently selected biological pathways. . *, **, and *** denote significance determined by Wilcoxon tests for p-values < 0.05, 0.01, and 0.001 respectively. The color indicates which experiment had significantly higher performance.

C
Figure S2. scMKL provides scalable, flexible, and accurate predictions across diverse datasets and identifies biologically relevant mechanisms through pathway/TF selection
c. Comparison of ATAC data groupings and transformations: Predictive performance for binary and TF-IDF normalized ATAC peaks using Hallmark gene sets, Cistrome and JASPAR TFBS as prior information. *, **, and *** denote significance determined by Wilcoxon tests for p-values < 0.05, 0.01, and 0.001 indicating which ATAC data transformation had significantly improved performance.

Figure S4. Biologically informed multimodal integration outperforms single-modality and state-of-the-art methods across diverse datasets a. Multimodal integration of RNA and ATAC

using Hallmark gene set-informed gene groups and JASPAR TFBS-informed peak groups consistently outperforms state-of-the-art algorithms across all datasets. *, **, *** are used to denote statistical significance from Wilcoxon test p values < 0.05, 0.01, and 0.001 respectively. Gold indicates scMKL had significantly better performance; black indicates the benchmark algorithm had significantly higher performance. **b.** Multimodal integration of RNA and ATAC using JASPAR-informed features not only surpasses single-modality predictions but also outperforms Hallmark-based multimodal integration across all datasets and evaluation metrics.

a

c

Figure S5. Hallmark pathways and Cistrome TFs in the RNA+ATAC multimodal setting.

a. Hallmark pathways as prior information. Comparison of classification performance (AUROC, F1-score, precision, recall) between RNA (red), ATAC (blue), and RNA+ATAC (purple) models across MCF-7, T-47D, and SLL multimodal datasets. *, **, and *** denote significance determined by Wilcoxon tests for p-values < 0.05, 0.01, and 0.001 indicating which experiments resulted had significantly improved performance. **c. Hallmark pathways and Cistrome TF as prior information.** Comparison of classification performance (AUROC, F1-score, precision, recall) between RNA (red), ATAC (blue), and RNA+ATAC (purple) models across MCF-7 and T-47D multimodal datasets. . *, **, and *** denote significance determined by Wilcoxon tests for p-values < 0.05, 0.01, and 0.001 respectively. The color indicates which experiment had significantly higher performance.

In conclusion, while the paper presents an innovative single-cell multiomics analysis method and demonstrates its strengths in RNA + ATAC data integration, there are several areas that need further improvement, particularly regarding interpretability, computational efficiency, generalization, and preprocessing strategy comparisons. The authors are encouraged to address these concerns in greater detail and provide additional experimental data to strengthen the scientific rigor and application potential of their approach.

We thank the reviewer for your thoughtful summary and constructive feedback. We appreciate your recognition of the innovation in our single-cell multiomics analysis method. In response to your concerns, we have taken steps to further address the key areas of interpretability, computational efficiency, generalization, and preprocessing strategy comparisons. Specifically:

1. **Interpretability:** We have expanded the discussion on how scMKL integrates multi-modal data, highlighting the use of biologically informed kernels and adding visualizations (e.g., heatmaps and dotplots) to illustrate feature importance and model transparency.
2. **Computational Efficiency:** We have provided additional comparisons of different preprocessing strategies and their impact on computational efficiency, particularly regarding ATAC data. This includes a detailed analysis of feature selection approaches and their effect on model performance under resource constraints. We provided the code for the scalability comparisons at https://github.com/ohsu-cedar-comp-hub/scMKL_paper in addition to scMKL's github at <https://github.com/ohsu-cedar-comp-hub/scMKL>.
3. **Generalization:** To address the model's generalization capabilities, we have added further validation experiments across independent datasets, including cross-cohort and cross-tumor type analysis. Our results demonstrate the robustness of scMKL in generalizing across lung cancer subtypes.
4. **Preprocessing Strategy Comparisons:** We conducted experiments to compare various preprocessing pipelines and presented their influence on scMKL's performance.

We believe these updates strengthen the manuscript. We thank the reviewer for their comments in improving these aspects and ensuring a clearer and more robust presentation of our approach.

- 1 Garcia-Martinez, L., Zhang, Y., Nakata, Y., Chan, H. L. & Morey, L. Epigenetic mechanisms in breast cancer therapy and resistance. *Nat Commun* **12**, 1786, doi:10.1038/s41467-021-22024-3 (2021).
- 2 Bredikhin, D., Kats, I. & Stegle, O. MUON: multimodal omics analysis framework. *Genome Biol* **23**, 42, doi:10.1186/s13059-021-02577-8 (2022).
- 3 ArulJothi, K. N. *et al.* Implications of reactive oxygen species in lung cancer and exploiting it for therapeutic interventions. *Med Oncol* **40**, 43, doi:10.1007/s12032-022-01900-y (2022).
- 4 Perillo, B. *et al.* ROS in cancer therapy: the bright side of the moon. *Exp Mol Med* **52**, 192-203, doi:10.1038/s12276-020-0384-2 (2020).
- 5 Jia, S. *et al.* Transcriptome Based Estrogen Related Genes Biomarkers for Diagnosis and Prognosis in Non-small Cell Lung Cancer. *Front Genet* **12**, 666396, doi:10.3389/fgene.2021.666396 (2021).
- 6 Wang, Y. *et al.* Immune characteristics analysis and construction of a four-gene prognostic signature for lung adenocarcinoma based on estrogen reactivity. *Bmc Cancer* **23**, 1047, doi:10.1186/s12885-023-11415-y (2023).
- 7 Shen, Y., Chen, J. Q. & Li, X. P. Differences between lung adenocarcinoma and lung squamous cell carcinoma: Driver genes, therapeutic targets, and clinical efficacy. *Genes Dis* **12**, 101374, doi:10.1016/j.gendis.2024.101374 (2025).
- 8 Acker, F. *et al.* KRAS Mutations in Squamous Cell Carcinomas of the Lung. *Front Oncol* **11**, 788084, doi:10.3389/fonc.2021.788084 (2021).
- 9 Quintanal-Villalonga, A. *et al.* Comprehensive molecular characterization of lung tumors implicates AKT and MYC signaling in adenocarcinoma to squamous cell transdifferentiation. *J Hematol Oncol* **14**, 170, doi:10.1186/s13045-021-01186-z (2021).
- 10 Li, F. *et al.* Comprehensive analysis of the role of a four-gene signature based on EMT in the prognosis, immunity, and treatment of lung squamous cell carcinoma. *Aging (Albany NY)* **15**, 6865-6893, doi:10.18632/aging.204878 (2023).